# EditGAN: High-Precision Semantic Image Editing

**Huan Ling**[1,2,3,*]    **Karsten Kreis**[1,*]    **Daiqing Li**[1]

**Seung Wook Kim**[1,2,3]    **Antonio Torralba**[4]    **Sanja Fidler**[1,2,3]

[1]NVIDIA    [2]University of Toronto    [3]Vector Institute    [4]MIT

{huling,kkreis,daiqingl,seungwookk,sfidler}@nvidia.com, torralba@mit.edu

## Abstract

Generative adversarial networks (GANs) have recently found applications in image editing. However, most GAN-based image editing methods often require large-scale datasets with semantic segmentation annotations for training, only provide high level control, or merely interpolate between different images. Here, we propose *EditGAN*, a novel method for high-quality, high-precision semantic image editing, allowing users to edit images by modifying their highly detailed part segmentation masks, e.g., drawing a new mask for the headlight of a car. EditGAN builds on a GAN framework that jointly models images and their semantic segmentations [1, 2], requiring only a handful of labeled examples – making it a scalable tool for editing. Specifically, we embed an image into the GAN's latent space and perform conditional latent code optimization according to the segmentation edit, which effectively also modifies the image. To amortize optimization, we find "editing vectors" in latent space that realize the edits. The framework allows us to learn an arbitrary number of editing vectors, which can then be directly applied on other images at interactive rates. We experimentally show that EditGAN can manipulate images with an unprecedented level of detail and freedom, while preserving full image quality.We can also easily combine multiple edits and perform plausible edits beyond EditGAN's training data. We demonstrate EditGAN on a wide variety of image types and quantitatively outperform several previous editing methods on standard editing benchmark tasks. Project page: https://nv-tlabs.github.io/editGAN.

## 1   Introduction

AI-driven photo and image editing has the potential to streamline the workflow of photographers and content creators and to enable new levels of creativity and digital artistry [3]. AI-based image editing tools have already found their way into consumer software in the form of neural photo editing filters, and the deep learning

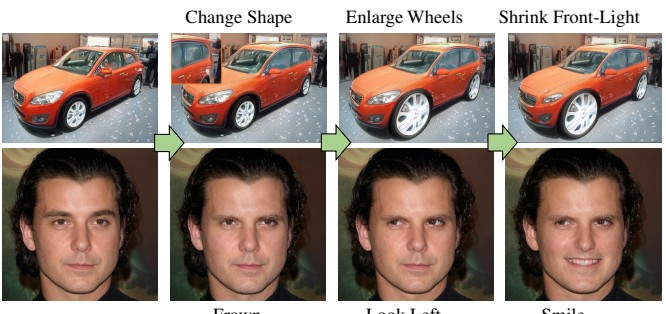

Figure 1: High-precision semantic image editing with EditGAN.

research community is actively developing further techniques. A particularly promising line of research uses generative adversarial networks (GANs) [4, 5, 6, 7, 8] and either embeds images into

---

*These authors contributed equally.

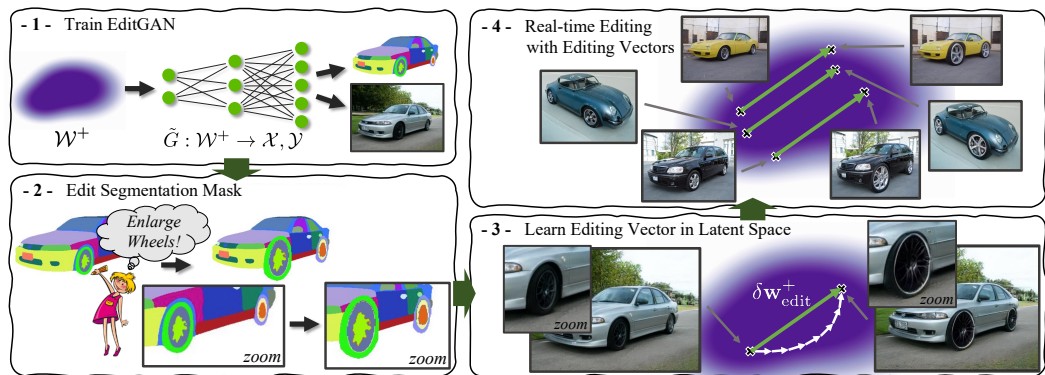

Figure 2: (**1**) EditGAN builds on a GAN framework that jointly models images and their semantic segmentations. (**2 & 3**) Users can modify segmentation masks, based on which we perform optimization in the GAN's latent space to realize the edit. (**4**) Users can perform editing simply by applying previously learnt editing vectors and manipulate images at interactive rates.

the GAN's latent space or works directly with GAN-generated images. Careful modifications of the latent embeddings then translate to desired changes in generated output, allowing, for example, to coherently change facial expressions in portraits [9, 10, 11, 12, 13, 14, 15, 16], change viewpoint or shapes and textures of cars [17], or to interpolate between different images in a semantically meaningful manner [18, 19, 20, 21].

Most GAN-based image editing methods fall into few categories. Some works rely on GANs conditioning on class labels or pixel-wise semantic segmentation annotations [19, 10, 22, 11], where different conditionings lead to modifications in the output, while others use auxiliary attribute classifiers [23, 15] to guide synthesis and edit images. However, training such conditional GANs or external classifiers requires large labeled datasets. Therefore, these methods are currently limited to image types for which large annotated datasets are available, like portraits [10]. Furthermore, even if annotations are available, most techniques offer only limited editing control, since these annotations usually consist only of high-level global attributes or relatively coarse pixel-wise segmentations. Another line of work focuses on mixing and interpolating features from different images [18, 19, 20, 21], thereby requiring reference images as editing targets and usually also not offering fine control. Other approaches carefully analyze and dissect GANs' latent spaces, finding disentangled latent variables suitable for editing [24, 25, 12, 13, 14, 26, 27], or control the GANs' network parameters [25, 28, 16]. Usually, these methods do not enable detailed editing and are often slow.

In this work, we are addressing these limitations and propose *EditGAN*, a novel GAN-based image editing framework that enables high-precision semantic image editing by allowing users to modify detailed object part segmentations. EditGAN builds on a recently proposed GAN that jointly models both images and their semantic segmentations based on the same underlying latent code [1, 2], and requires as few as 16 labeled examples – allowing it to scale to many object classes and choices of part labels. We achieve editing by modifying the segmentation mask according to a desired edit and optimizing the latent code to be consistent with the new segmentation, thus effectively changing the RGB image. To achieve efficiency, we learn *editing vectors* in latent space that realize the edits, and that can be directly applied on other images, without any or only few additional optimization steps. We can thus pre-train a library of interesting edits that a user can directly utilize in an interactive tool.

We apply EditGAN on a wide range of images, including images of cars, cats, birds, and human faces, demonstrating unprecedented high-precision editing. We perform quantitative comparisons to multiple baselines and outperform them in metrics such as identity preservation, quality preservation, and target attribute accuracy, while requiring orders of magnitude less annotated training data. EditGAN is the first GAN-driven image editing framework, which simultaneously (**i**) offers very high-precision editing, (**ii**) requires only very little annotated training data (and does not rely on external classifiers), (**iii**) can be run interactively in real time, (**iv**) allows for straightforward compositionality of multiple edits, (**v**) and works on real embedded, GAN-generated, and even out-of-domain images.

## 2 Related Work

**Image Editing and Manipulation.** Image Editing has a long history in computer vision and graphics, as well as machine learning [29, 30, 31, 32, 33, 34, 35, 36, 37, 38, 39, 40, 18, 11, 28, 41, 42, 16]. Recently, deep generative models [4, 43, 44], in particular modern GANs [6, 45, 7, 46, 8], received

much attention as a promising tool for efficient image editing, as it was found that latent space manipulations often lead to interpretable and predictable changes in output [47, 24, 48, 49, 26, 27, 50].

GAN-based image editing methods can be broadly sorted into a number of categories. **(i)** One line of work relies on the careful dissection of the GAN's latent space, aiming to find interpretable and disentangled latent variables, which can be leveraged for image editing, in a fully unsupervised manner [47, 24, 25, 12, 13, 14, 48, 49, 26, 27, 50, 51]. Although powerful, these approaches usually do not result in any high-precision editing capabilities. The editing vectors we are learning in EditGAN would be too hard to find independently without segmentation-based guidance. **(ii)** Other works utilize GANs that condition on class or pixel-wise semantic segmentation labels to control synthesis and achieve editing [9, 52, 46, 19, 10, 22, 11]. Hence, these works usually rely on large annotated datasets, which are often not available, and even if available, the possible editing operations are tied to whatever labels are available. This stands in stark contrast to EditGAN, which can be trained in a semi-supervised fashion with very little labeled data and where an arbitrary number of high-precision edits can be learnt. **(iii)** Furthermore, auxiliary attribute classifiers have been used for image manipulation [23, 15], thereby still relying on annotated data and usually only providing high-level control. **(iv)** Image editing is often explored in the context of "interpolating" between a target and different reference image in sophisticated ways, for example by replacing certain features in a given image with features from a reference images [18, 19, 20, 21]. From the general image editing perspective, the requirement of reference images limits the broad applicability of these techniques and prevents the user from performing specific, detailed edits for which potentially no reference images are available. **(v)** Recently, different works proposed to directly operate in the parameter space of the GAN instead of the latent space to realize different edits [25, 28, 16]. For example, [25, 28] essentially specialize the generator network for certain images at test time to aid image embedding or "rewrite" the network to achieve desired semantic changes in output. The drawback is that such specializations prevent the model from being used in real-time on different images and with different edits. [16] proposed an approach that more directly analyses the parameter space of a GAN and treats it as a latent space in which to apply edits. However, the method still merely discovers edits in the network's parameter space, rather than actively defining them like we do. It remains unclear whether their method can combine multiple such edits, as we can, considering that they change the GAN parameters themselves. **(vi)** Finally, another line of research targets primarily very high-level image and photo stylization and global appearance modifications [37, 53, 54, 55, 52, 56, 46, 57, 41].

Generally, most works only do relatively high-level and not the detailed, high-precision editing, which EditGAN targets. Hence, we consider EditGAN as complementary to this body of work.

**GANs and Latent Space Image Embedding.** EditGAN builds on top of DatasetGAN [1] and SemanticGAN [2], which proposed to jointly model images and their semantic segmentations using shared latent codes. However, these works leveraged this model design only for semi-supervised learning, not for editing. EditGAN also relies on an encoder, together with optimization, to embed new images to be edited into the GAN's latent space. This task in itself has been studied extensively in different contexts before, and we are building on these works. Previous papers studied encoder-based methods [58, 59, 60, 61, 62], used primarily optimization-based techniques [63, 64, 65, 66, 67, 68, 69, 26], and developed hybrid approaches [63, 24, 25, 70, 71].

Finally, a concurrent paper [72] shares similarities with DatasetGAN [1], on which our method builds, and explores an editing approach related to our EditGAN as one of its applications. However, our editing approach is methodologically different and leverages editing vectors, and also demonstrates significantly more diverse and stronger experimental results. Furthermore, [73] shares some high-level ideas with EditGAN; however, it leverages the CLIP [74] model and targets text-driven editing.

## 3 High-Precision Semantic Image Editing with EditGAN

### 3.1 Background

EditGAN's image generation component is StyleGAN2 [7, 8], currently the state-of-the-art GAN for image synthesis. The StyleGAN2 generator maps latent codes $\mathbf{z} \in \mathcal{Z}$, drawn from a multivariate Normal distribution, into realistic images. A latent code $\mathbf{z}$ is first transformed into an intermediate code $\mathbf{w} \in \mathcal{W}$ by a non-linear mapping function and then further transformed into $K + 1$ vectors, $\mathbf{w}^0, ..., \mathbf{w}^K$, through learned affine transformations. These transformed latent codes are fed into synthesis blocks, whose outputs are deep feature maps.

Deep generative models such as StyleGAN2, which are trained to synthesize highly realistic images, acquire a semantic understanding of the modeled images in their high-dimensional feature space. Recently, DatasetGAN [1] and SemanticGAN [2] built on this insight to learn a joint distribution $p(\mathbf{x}, \mathbf{y})$ over images $\mathbf{x}$ and pixel-wise semantic segmentation labels $\mathbf{y}$, while requiring only a handful of labeled examples. EditGAN utilizes this joint distribution $p(\mathbf{x}, \mathbf{y})$ to perform high-precision semantic image editing of real and synthesized images.

Both methods [1, 2] model $p(\mathbf{x}, \mathbf{y})$ by adding an additional segmentation branch to the image generator, which is a pre-trained StyleGAN [1]. We follow DatasetGAN [1], which applies a simple three-layer multi-layer perceptron classifier on the layer-wise concatenated and appropriately upsampled feature maps. This classifier operates on the concatenated feature maps in a per-pixel fashion and predicts the segmentation label of each pixel.

### 3.2 Segmentation Training and Inference by Embedding Images into GAN's Latent Space

To both train the segmentation branch and perform segmentation on a new image, we embed an image into the GAN's latent space using an encoder and optimization. To this end, we build on previous works [66, 62, 2] and train an encoder that embeds images into $\mathcal{W}^+$ space, which is defined as $\mathcal{W}$ but where the $\mathbf{w}$'s are modeled independently [66, 62]. Our objectives to train this encoder consist of standard pixel-wise L2 and perceptual LPIPS reconstruction losses using both the real training data as well as samples from the GAN itself. For the GAN samples, we also explicitly regularize the encoder with the known underlying latent codes. In practice, we use the encoder to initialize images' latent space embeddings and then iteratively refine the latent code $\mathbf{w}^+$ via optimization, again using standard reconstruction objectives.

In that way, we embed the annotated images $\mathbf{x}$ from a dataset labeled with semantic segmentations into latent space, and train the segmentation branch of the generator using standard supervised learning objectives, i.e., the cross entropy loss. We keep the image generator's weights frozen and only backpropagate the loss to the segmentation branch [1]. After training the segmentation branch, we can formally define a generator $\tilde{G} : \mathcal{W}^+ \rightarrow \mathcal{X}, \mathcal{Y}$ that models the joint distribution $p(\mathbf{x}, \mathbf{y})$ of images $\mathbf{x}$ and semantic segmentations $\mathbf{y}$. Details about encoder and segmentation branch training as well as optimization for image embedding can be found in the Appendix.

### 3.3 Finding Semantics in Latent Space via Segmentation Editing

The key idea of EditGAN lies in leveraging the joint distribution $p(\mathbf{x}, \mathbf{y})$ of images and semantic segmentations for high-precision image editing. Given a new image $\mathbf{x}$ to be edited, we can embed it into EditGAN's $\mathcal{W}^+$ latent space, as described above (alternatively, we can also sample images from the model itself and use those). The segmentation branch will then generate the corresponding segmentation $\mathbf{y}$, since segmentations and RGB im-

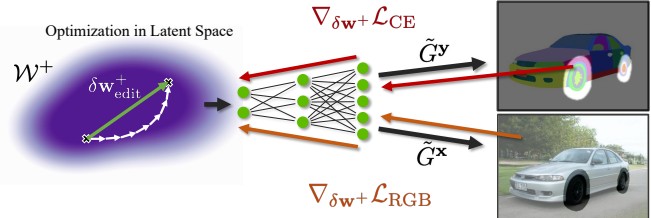

Figure 3: We modify semantic segmentations and optimize the shared latent code for consistency with the new segmentation *within* the editing region, and with the RGB appearance *outside* the editing region. Corresponding gradients are backpropagated through the shared generator. The result is a latent space editing vector $\delta\mathbf{w}^+_{\text{edit}}$.

ages share the same latent codes $\mathbf{w}^+$. Using simple interactive digital painting or labeling tools, we can now manually modify the segmentation according to a desired edit. We denote the edited segmentation mask by $\mathbf{y}_{\text{edited}}$. Starting from the embedding $\mathbf{w}^+$ of the unedited image $\mathbf{x}$ and segmentation $\mathbf{y}$, we can then perform optimization within $\mathcal{W}^+$ to find a new $\mathbf{w}^+_{\text{edited}} = \mathbf{w}^+ + \delta\mathbf{w}^+_{\text{edit}}$ consistent with the new segmentation $\mathbf{y}_{\text{edited}}$, while allowing the RGB output $\mathbf{x}$ to change within the editing region.

Formally, we are seeking an *editing vector* $\delta\mathbf{w}^+_{\text{edit}} \in \mathcal{W}^+$ such that $(\mathbf{x}_{\text{edited}}, \mathbf{y}_{\text{edited}}) = \tilde{G}(\mathbf{w}^+ + \delta\mathbf{w}^+_{\text{edit}})$, where $\tilde{G}$ denotes the fixed generator that synthesizes both images and segmentations. Defining $(\mathbf{x}', \mathbf{y}') = \tilde{G}(\mathbf{w}^+ + \delta\mathbf{w}^+)$, we perform optimization to approximate $\delta\mathbf{w}^+_{\text{edit}}$ by $\delta\mathbf{w}^+$. The region of interest $r$ within which we expect the image to change due to the edit is formally given by

$$r = \left\{ p : c_p^{\mathbf{y}} \in Q_{\text{edit}} \right\} \cup \left\{ p : c_p^{\mathbf{y}_{\text{edited}}} \in Q_{\text{edit}} \right\} \tag{1}$$

which means that $r$ is defined by all pixels $p$ whose part segmentation labels $c_p^{\{\mathbf{y}, \mathbf{y}_{\text{edited}}\}}$ according to either the initial segmentation $\mathbf{y}$ or the edited one $\mathbf{y}_{\text{edited}}$ are within an edit-specific pre-specified list

$Q_{\text{edit}}$ of part labels relevant for the edit. For example, when modifying the wheel in a photo of a car $Q_{\text{edit}}$ would contain all part labels related to the wheels, such as tire, spoke, and wheelhub (see Fig. 3). We use a further buffer of 5 pixels to give the GAN freedom in modeling the transition between the edited and non-edited area. In practice, $r$ acts as a binary pixel-wise mask (see Eqs. 2 and 3 below).

Note that $\mathbf{x}_{\text{edited}}$ is not available during optimization. After all, $\mathbf{x}_{\text{edited}}$ is the edited image we are ultimately intested in. It emerges indirectly when optimizing for the segmentation modification, since images and segmentations are closely tied together in the joint distribution $p(\mathbf{x}, \mathbf{y})$ modeled by $\tilde{G}$. We further define $\mathbf{x}' = \tilde{G}^{\mathbf{x}}(\mathbf{w}^+ + \delta\mathbf{w}^+)$ as $\tilde{G}$'s image generation and $\mathbf{y}' = \tilde{G}^{\mathbf{y}}(\mathbf{w}^+ + \delta\mathbf{w}^+)$ as $\tilde{G}$'s segmentation generation branch.

To find $\delta\mathbf{w}^+$, approximating $\delta\mathbf{w}^+_{\text{edit}}$, we use the following losses as minimization targets:

$$\mathcal{L}_{\text{RGB}}(\delta\mathbf{w}^+) = L_{\text{LPIPS}}(\tilde{G}^{\mathbf{x}}(\mathbf{w}^+ + \delta\mathbf{w}^+) \odot (1 - r), \ \mathbf{x} \odot (1 - r))$$
$$+ L_{L2}(\tilde{G}^{\mathbf{x}}(\mathbf{w}^+ + \delta\mathbf{w}^+) \odot (1 - r), \ \mathbf{x} \odot (1 - r)) \quad (2)$$

$$\mathcal{L}_{\text{CE}}(\delta\mathbf{w}^+) = H(\tilde{G}^{\mathbf{y}}(\mathbf{w}^+ + \delta\mathbf{w}^+) \odot r, \ \mathbf{y}_{\text{edited}} \odot r) \quad (3)$$

where $H$ denotes the pixel-wise cross-entropy, $L_{\text{LPIPS}}$ loss is based on the Learned Perceptual Image Patch Similarity (LPIPS) distance [75], and $L_{L2}$ is a regular pixel-wise L2 loss. $\mathcal{L}_{\text{RGB}}(\delta\mathbf{w}^+)$ ensures that the image appearance does not change *outside* the region of interest, while $\mathcal{L}_{\text{CE}}(\delta\mathbf{w}^+)$ ensures that the target segmentation $\mathbf{y}_{\text{edited}}$ is enforced *within* the editing region (see visualization in Fig. 3). When editing human faces, we also apply the identity loss [62]:

$$\mathcal{L}_{\text{ID}}(\delta\mathbf{w}^+) = \langle R(\tilde{G}^{\mathbf{x}}(\mathbf{w}^+ + \delta\mathbf{w}^+)), R(\mathbf{x}) \rangle \quad (4)$$

with $R$ denoting the pretrained ArcFace feature extraction network [76] and $\langle \cdot, \cdot \rangle$ cosine-similiarity.

The final objective function for optimization then becomes:

$$\mathcal{L}_{\text{editing}}(\delta\mathbf{w}^+) = \lambda_1^{\text{editing}}\mathcal{L}_{\text{RGB}}(\delta\mathbf{w}^+) + \lambda_2^{\text{editing}}\mathcal{L}_{\text{CE}}(\delta\mathbf{w}^+) + \lambda_3^{\text{editing}}\mathcal{L}_{\text{ID}}(\delta\mathbf{w}^+) \quad (5)$$

with hyperparameters $\lambda_{1,...,3}^{\text{editing}}$. The only "learnable" variable is the editing vector $\delta\mathbf{w}^+$; all neural networks are kept fixed. After optimizing $\delta\mathbf{w}^+$ with the objective function, we can use $\delta\mathbf{w}^+ \approx \delta\mathbf{w}^+_{\text{edit}}$. Note that there is a certain amount of ambiguity in how the segmentation modification is realized in RGB output. We rely on the GAN generator, trained to synthesize realistic images, to modify the RGB values in the editing region in a plausible way consistent with the segmentation edit.

### 3.4 Different Ways of Editing during Inference

The latent space editing vectors $\delta\mathbf{w}^+_{\text{edit}}$ obtained by optimization as described are semantically meaningful and often disentangled with other attributes. Therefore, for new images $\mathbf{x}$ to be edited, we can embed the images into the $\mathcal{W}^+$ latent space and the same editing operations can be directly performed by applying the previously learnt $\delta\mathbf{w}^+_{\text{edit}}$ as $(\mathbf{x}', \mathbf{y}') = G(\mathbf{w}^+ + s_{\text{edit}}\delta\mathbf{w}^+_{\text{edit}})$ without doing any optimization from scratch again. In other words, the learnt editing vectors $\delta\mathbf{w}^+$ amortize the iterative optimization that was necessary to achieve the edit initially. For well-disentangled editing operations, $\mathbf{x}'$ can be used directly as the edited image $\mathbf{x}_{\text{edited}}$. Note that we introduced $s_{\text{edit}}$, a scalar editing coefficient, which effectively scales and controls the editing magnitude during inference. For $s_{\text{edit}} = 0$, we do not do any editing at all, while for $s_{\text{edit}} > 1$ we manipulate the images with an effectively larger editing operation in latent space, leading to exaggerated effects.

Unfortunately, disentanglement is not always perfect and the editing vectors $\delta\mathbf{w}^+_{\text{edit}}$ do not always translate perfectly to other images. We can remove editing artifacts in other regions of the image by a few additional optimization steps at test time. Specifically, we can use the exact same minimization objectives as above, using the initial prediction $\mathbf{y}'$, obtained after applying the editing vector $\delta\mathbf{w}^+_{\text{edit}}$, as $\mathbf{y}_{\text{edited}}$. This assumes that the editing vector still induces a plausible segmentation change when applied on other images and that artifacts only arise in RGB output. The RGB objective $\mathcal{L}_{\text{RGB}}$ then removes these editing artifacts outside the editing region, while $\mathcal{L}_{\text{CE}}$ ensures that the modified segmentation stays as predicted by the editing vector.

Summarizing, we can perform image editing with EditGAN in three different modes:

- **Real-time Editing with Editing Vectors.** For localized, well-disentangled edits we perform editing purely by applying previously learnt editing vectors with varying scales $s_{\text{edit}}$ and manipulate images at interactive rates.

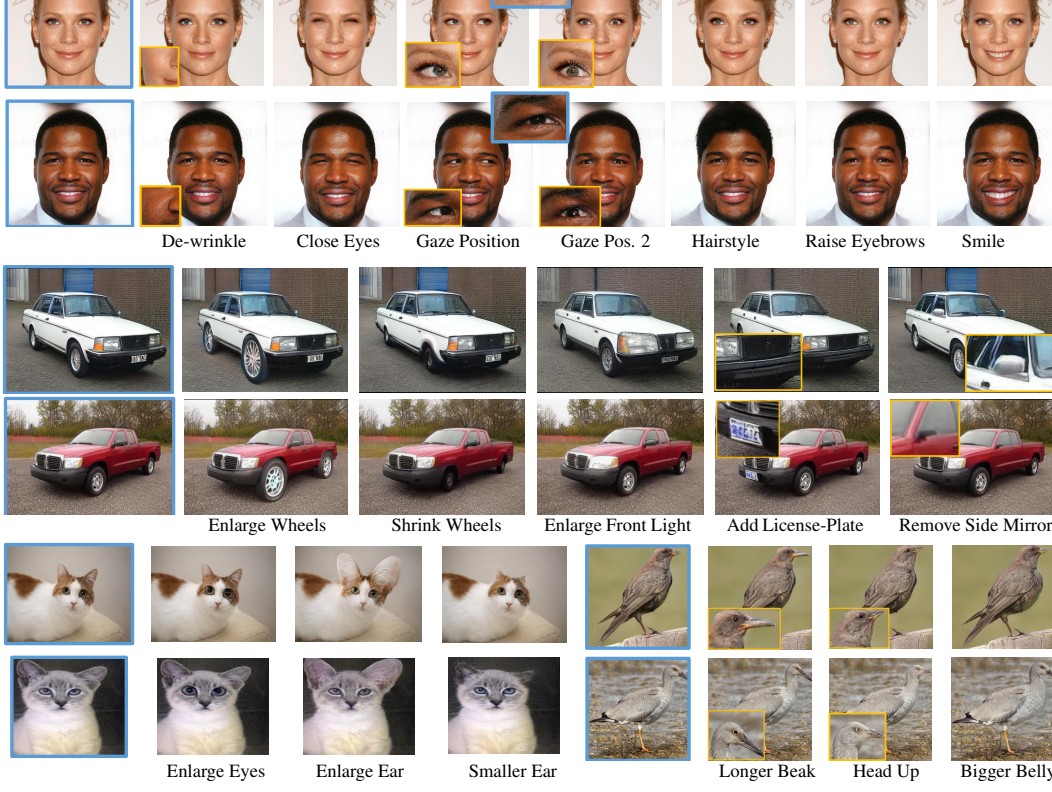

Figure 4: Examples of segmentation-driven edits with EditGAN. Results are based on editing with editing vectors and 30 steps self-supervised refinement. *Blue boxes*: Original images. *Orange boxes*: Zoom-in views.

- **Vector-based Editing with Self-Supervised Refinement.** For localized edits that are not perfectly disentangled with other parts of the image, we can remove editing artifacts by additional optimization at test time, while initializing the edit using the learnt editing vectors.
- **Optimization-based Editing.** Image-specific and very large edits do not transfer to other images via editing vectors. For such operations, we perform optimization from scratch.

## 4 Experiments

We extensively evaluate EditGAN on images across four different categories: Cars ($384{\times}512$ spatial resolution), Birds ($512{\times}512$), Cats ($256{\times}256$), and Faces ($1024{\times}1024$).

**Implementation** We train our segmentation branch as described in Sec. 3.2 using 16, 16, 30, and 30 image-mask pairs as labeled training data for Faces, Cars, Birds, and Cats, respectively. We utilize very highly-detailed part segmentations from [1]. The annotation scheme for faces is shown in Fig. 7, all others are presented in the Appendix. When editing is done purely optimization-based or when learning the editing vectors, we always perform 100 steps of optimization using Adam [77]. For Car, Cat, and Faces, we use real images from DatasetGAN's test set that were not part of GAN training to demonstrate editing functionality. These images are first embedded into EditGAN's latent space via an encoder and optimization as described in Sec. 3.2. For Birds, we show editing on GAN-generated images. Model details and hyperparameters are provided in the Appendix.

### 4.1 Qualitative Results

**In-Domain Results** In Fig. 4, we demonstrate our EditGAN framework when applying previously learnt editing vectors $\delta\mathbf{w}^{+}_{\text{edit}}$ on novel images and refining with 30 steps of optimization. Our editing operations preserve high image quality and are well disentangled for all classes. We also show the ability to combine multiple different edits in Fig. 5. To the best of our knowledge, no previous methods can perform as complex and high-precision edits as we do, while preserving image quality and subject identity. In Fig. 8, we demonstrate that we can even perform extremely high-precision

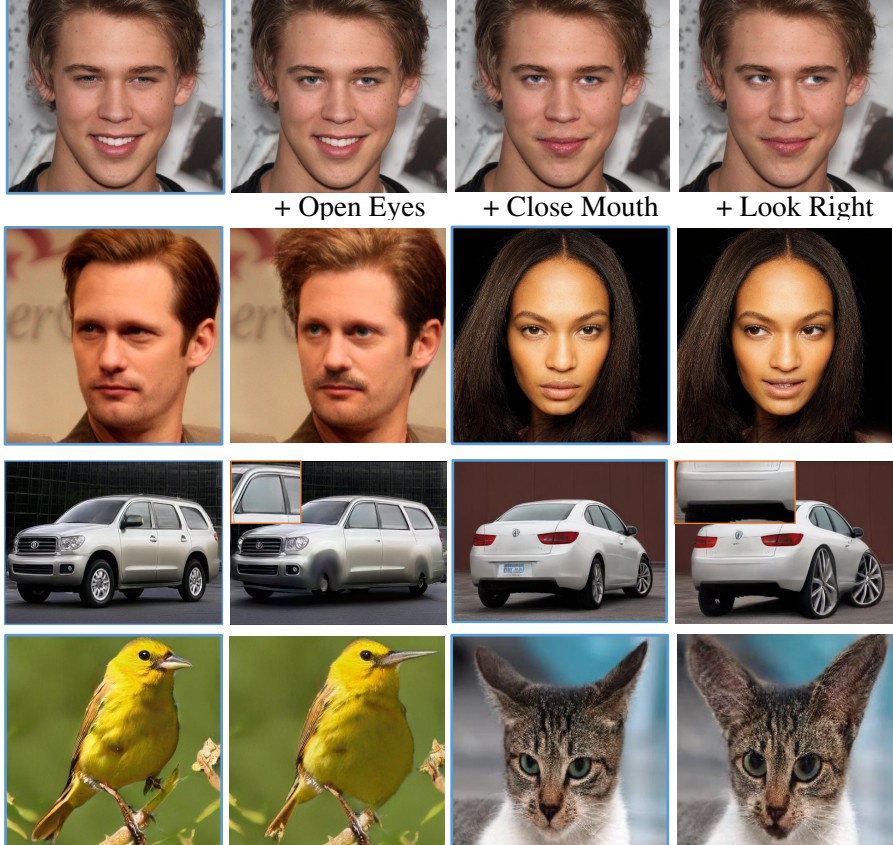

Figure 5: We combine multiple edits. Results are based on editing with editing vectors and 30 steps self-supervised refinement. *Blue boxes*: Original images. Edits in detail: *Second row, first person:* open eyes, add hair, add mustache. *Second person:* smile, look left. *Third row, first car:* remove mirror, remove door handle, shrink wheels. *Second car:* remove license plate, enlarge wheels. *Third row, bird:* longer beak, bigger belly, head up. *Third row, cat:* open mouth, bigger ear, bigger eyes.

edits, such as rotating a car's wheel spoke or dilating pupils. EditGAN can edit semantic parts of objects that consist of only few pixels. At the same time, we can use EditGAN to perform large-scale modifications, too: In Fig. 9, we present how we can remove the entire roof of a car or convert it to a station wagon-like vehicle, simply by modifying the segmentation mask accordingly and optimizing. It is worth noting that several of our editing operations generate plausible manipulated images unlike those appearing in the GAN training data. For example, the training data does not include cats with overly large eyes or ears. Nevertheless, we achieve such edits in a high-quality manner.

The edits in Figs. 4, 5 and 8 are based on learnt editing vectors with self-supervised refinement. However, without such refinement usually only very minor artifacts occur, as shown in Fig. 10, hence allowing for real-time high-precision semantic image editing (discussed in detail below).

**Out-of-Domain Results** We demonstrate the generalization capability of EditGAN to out-of-domain data on the MetFaces [8] data set. We use our EditGAN model trained on FFHQ [8], and create editing vectors $\delta \mathbf{w}^{+}_{\text{edit}}$ using in-domain real faces. We then embed out-of-domain MetFaces partraits (with 100 steps optimization) and apply the editing vectors with 30 steps self-supervised refinement. The results are shown in Fig. 6. We find that our editing operations seamlessly translate even to such far out-of-domain examples.

## 4.2 Quantitative Results

To quantitatively measure EditGAN's image editing capabilities, we use the smile edit benchmark introduced by MaskGAN [10]. Faces with neutral expressions are converted into smiling faces and performance is measured by three metrics: **a. Semantic Correctness:** Using a pre-trained smile attribute classifier, we measure whether the faces show smiling expressions after editing. **b. Distribution-level Image Quality:** Frechet Inception Distance (FID) [78, 79] and Kernel Inception Distance (KID) [80] are calculated between 400 edited test images and the CelebA-HD test dataset. **c.**

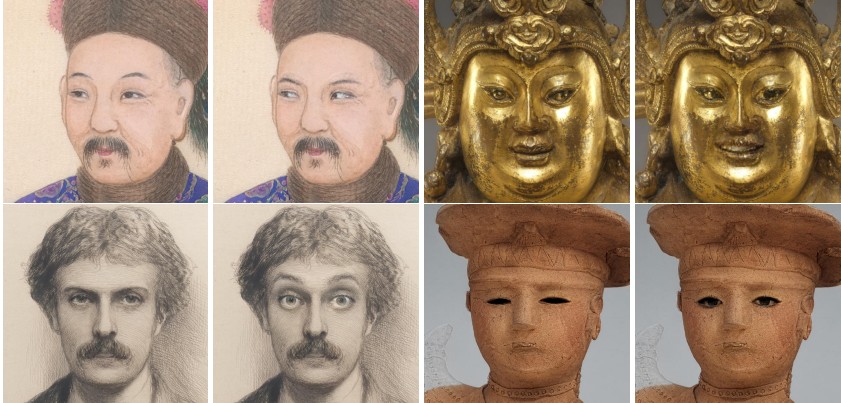

Figure 6: We combine multiple edits on out-of-domain images. Results are based on editing with editing vectors and 30 steps self-supervised refinement. Edits in detail: *First row, first example:* look left, frown. *Second example:* smile, look right. *Second row, first example:* open eyes, lift eyebrow. *Second example:* open eyes.

**Identity Preservation:** Using the pretrained ArcFace feature extraction network [76], we measure whether the subjects' identity is maintained when applying the edit. Specifically, we report cosine-similiarity between original and edited images. Further details can be found in the Appendix.

For our EditGAN, we simply learn a smiling editing vector $\delta \mathbf{w}^+_{edit}$ using a hold-out neutral expression face image. We embed it into EditGAN, infer its pixel-wise segmentation labels, and manually modify the segmentation towards a smile. Then we perform optimization in latent space, as described above, to learn the editing vector. For the results in Tab. 1, it is applied with unit scale $s_{edit}=1$ on new images. We do

| Metric | # Mask Annot. | # Attribute Annot. | Attribute Acc.(%) ↑ | FID ↓ | KID ↓ | ID Score ↑ |
|---|---|---|---|---|---|---|
| MaskGAN [10] | 30,000 | - | 77.3 | 46.84 | 0.020 | 0.4611 |
| LocalEditing [18] | - | - | 26.0 | 41.26 | 0.012 | 0.5823 |
| LocalEditing - Encoding4Editing [81] | - | - | 41.75 | 48.28 | 0.016 | 0.6603 |
| InterFaceGAN [13] | - | 30,000 | 83.5 | **39.42** | **0.010** | 0.7295 |
| EditGAN (ours) | 16 | - | **91.5** | 41.74 | 0.013 | 0.7047 |
| EditGAN$^+$30 (ours) | 16 | - | 85.8 | 40.83 | 0.012 | **0.7452** |
| StyleGAN2 Distillation [82] | - | 30,000 | 98.3 | 45.09 | 0.013 | 0.7823 |

Table 1: Quantitative comparisons to multiple baselines on the smile edit benchmark.

not use the identity loss (Eq. 4) in this experiment, since identity preservation is already a target metric itself. We compare our method with three strong baselines: **(i)** *MaskGAN*[2] [10]: It takes non-smiling images, their segmentation masks, and a target smiling segmentation mask as inputs. Note that training MaskGAN requires large annotated datasets, in contrast to us. We also compare to **(ii)** *LocalEditing*[3] [18]: It clusters GAN features to achieve local editing and relies on reference images, in this case images of faces with smiling expressions. Another baseline we use is **(iii)** *InterFaceGAN*[4] [13]: Similar to EditGAN, InterFaceGAN aims at finding editing vectors in latent space. However, it uses auxiliary attribute classifiers, relies on large annotated datasets, and can generally not achieve the fine editing control of our EditGAN. Finally, we compare to **(iv)** *StyleGAN2 Distillation*[5] [82], which creates an alternative approach that does not require real image embeddings and also relies on an editing-vector model to create a training dataset.

Results are reported in Tab. 1. Using $1,875\times$ less training labels, we outperform MaskGAN on all three metrics. We similarly obtain significantly stronger results than LocalEditing. In our observation, LocalEditing does not work well on real image embeddings. We further exploit a better encoder [81] for the LocalEditing baseline, which leads to a significant improvement in attribute accuracy and ID score, but slightly worse FID & KID scores. We find that EditGAN outperforms InterFaceGAN on identity preservation and attribute classification accuracy, while InterFaceGAN reaches slightly better FID & KID scores (for the results in Tab. 1, the latent space edits learnt by InterfaceGAN are also applied with unit scale, like for EditGAN). In Fig. 11, we report a more detailed comparison to InterFaceGAN, where we apply the smile editing vectors with different scale coefficients from zero to two. As shown, when the editing vector scale is small, the identity score is high while the smiling attribute score is low, since the modification of the original images is minimal. We find that our real-time editing with editing vectors is on-par with InterFaceGAN. When we perform self-supervised

---

[2]https://github.com/switchablenorms/CelebAMask-HQ
[3]https://github.com/IVRL/GANLocalEditing
[4]https://github.com/genforce/interfacegan
[5]https://github.com/EvgenyKashin/stylegan2-distillation

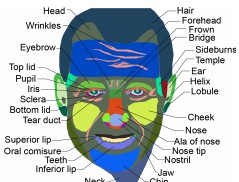

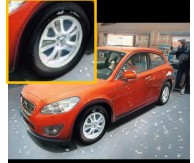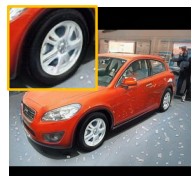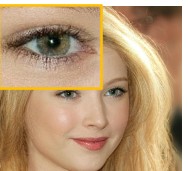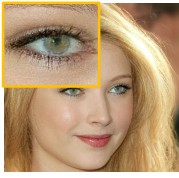

Figure 7: Face part labeling schema [1].

Figure 8: High-precision editing with EditGAN for extreme details. *Left:* We rotate the spoke. *Right:* We modify pupil size. Results are based on editing with editing vectors and 30 steps self-supervised refinement.

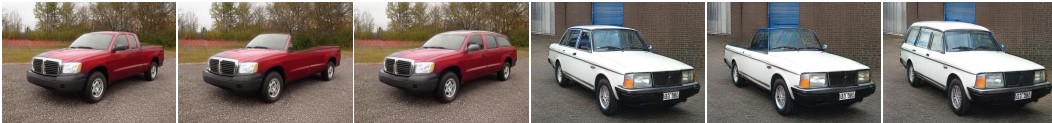

Figure 9: Pure optimization-based editing. We demonstrate large-scale semantic edits that do not transfer seamlessly to other images via editing vectors. Hence, we perform optimization from scratch.

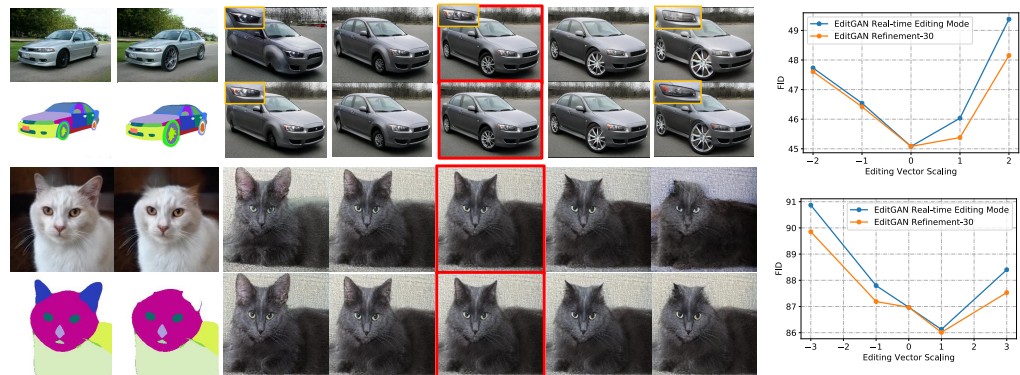

Figure 10: *Left:* We apply learnt editing vectors with varying scales (see 5 markers in FID plots) both without (top row for each class) and with (bottom row for each class) additional 30-step self-supervised refinement to correct artifacts. Red boxes denote original images. For each class, the leftmost image is the one used to learn the editing vector, with the editing result next to it and orginal and modified segmentations below. *Right:* Visual quality after editing with different scales as measured by FID with and without refinement.

refinement at test time, EditGAN outperforms InterFaceGAN. In Tab. 1, we also compare with StyleGAN2 Distillation [82], which achieves strong performance. However, StyleGAN2 Distillation relies on pre-trained classifiers, like InterfaceGAN, and only enables relatively high-level editing of image attributes for which large-scale annotations exit. Moreover, it distills edits into separate Pixel2PixelHD networks, such that a new network needs to be trained for each edit, limiting broad, user-interactive applicability. Hence, we consider StyleGAN2 Distillation orthogonal to our EditGAN.

**Running Time** We carefully measure the run time of our editing on an NVIDIA Tesla V100 GPU. Conditional optimization, given an edited segmentation mask, with 30 (60) optimization steps takes 11.4 (18.9) seconds. This operation provides us the editing vector. Application of editing vectors is almost instantaneous, taking only 0.4 seconds, therefore allowing for complex real-time interactive editing. A 10 (30) step self-supervised refinement would add an additional 4.2 (9.5) seconds.

### 4.3 Ablation Studies: Self-Supervised Refinement and Editing Vector Scale

Fig. 11 also contains a quantitative ablation study on the number of additional optimization steps done when initializing an edit with a learnt editing vector and refining with additional optimization. Generally, the more refinement steps we perform, the better the performance our model can achieve. As shown in Fig. 11, we find that further optimization can indeed slightly improve performance. Specifically, here we improve the trade-off between maintaining identity and achieving the desired semantic operation when performing editing with different scalings $s_{edit}$ of the editing vector. However, performing many steps of optimization leads to a run-time vs. performance trade-off, and our results suggest that the improvement beyond 30 additional optimization steps becomes marginal.

In Fig. 10, we analyze the editing vector scale and self-supervised refinement visually and with respect to perceptual metrics. As highlighted in the zoom-in areas, small artifacts can appear due

to imperfect disentanglement in latent space when applying editing operations with large scales. Self-supervised refinement successfully cleans these editing errors up. We also apply the same edit with different scales on 400 test images and measure FID with respect to 10,000 data from GAN training, inspired by the analyses in [16]. We can clearly see that image quality degrades as measured by FID, the stronger the edit is applied. We also observe small improvements with the iterative refinement on this metric, although the difference is small. Further details are in the Appendix. We conclude that for most editing operations, real-time editing without iterative refinement already performs very well. However, to clean up artifacts and maintain highest image quality possible, self-supervised refinement with a couple of additional optimization steps is always available.

Additional experiments are presented in the Appendix.

## 5 Conclusions

**Limitations** Like all GAN-based image editing methods, EditGAN is limited to images that can be modeled by the GAN. This makes EditGAN's application on, for instance, photos of vivid city scenes challenging. Although most of our high-precision edits readily transfer to other images via learnt editing vectors, we also encountered challenging edits that required iterative optimization on each example. Future research therefore includes speeding up the optimization for such edits as well as building better generative models with more disentangled latent spaces.

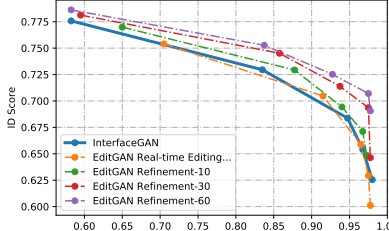

Figure 11: InterFaceGAN's and EditGAN's performance on the smile edit benchmark for different editing vector scalings (scale increases from top-left points towards bottom-right points; see main text and Appendix for details). For EditGAN, we optionally add 10, 30 or 60 additional optimization steps.

**Summary** We propose EditGAN, a novel method for high-precision, high-quality semantic image editing. It relies on a GAN that jointly models RGB images and their pixel-wise semantic segmentations and that requires only very few annotated data for training. Editing is achieved by performing optimization in latent space while conditioning on edited segmentation masks. This optimization can be amortized into editing vectors in latent space, which can be applied on other images directly, allowing for real-time interactive editing without any or only little further optimization. We demonstrate a broad variety of editing operations on different kinds of images, achieving an unprecedented level of flexibility and freedom in terms of editing, while preserving high image quality.

## 6 Broader Impact

Where previous generative modeling-based image editing methods offer only limited high-level editing capabilities, our method provides users unprecedented high-precision semantic editing possibilities. Our proposed techniques can be used for artistic purposes and creative expression and benefit designers, photographers, and content creators [3]. AI-driven image editing tools like ours promise to democratize high-quality image editing. Related methods have already found their way into everyday applications in the form of neural photo editing filters. On a larger scale, the ability to synthesize data with specific attributes can be leveraged in training and finetuning machine learning models.

At the same time, more precise photo editing also offers opportunities for advanced photo manipulation for nefarious purposes. The recent progress of generative models and AI-driven photo editing has profound implications on image authenticity and beyond, which is an area of active debate [83]. As one potential way to tackle these challenges, methods for automatically validating real images and detecting manipulated or fake images are being developed by the research community [84, 85]. Furthermore, generative models like ours are usually only as good as the data they were trained on. Therefore, biases in the underlying datasets are still present in the synthesized images and preserved even when applying our proposed editing methods. It is therefore important to be aware of such biases in the underlying data and counteract them, for example by actively collecting more representative data or by using bias correction methods, an area of active research [86, 87, 88, 89].

## Funding Statement

This work was funded by NVIDIA. Huan Ling and Seung Wook Kim acknowledge additional revenue in the form of student scholarships from University of Toronto and the Vector Institute, which are not in direct support of this work.

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
