# Supplementary Material:
# EditGAN: High-Precision Semantic Image Editing

**Huan Ling**[1,2,3,*]     **Karsten Kreis**[1,*]     **Daiqing Li**[1]

**Seung Wook Kim**[1,2,3]     **Antonio Torralba**[4]     **Sanja Fidler**[1,2,3]

[1]NVIDIA    [2]University of Toronto    [3]Vector Institute    [4]MIT

{huling,kkreis,daiqingl,seungwookk,sfidler}@nvidia.com, torralba@mit.edu

## 1  Model and Training Details

We first provide additional details about our EditGAN.

### 1.1  Image GAN

EditGAN uses StyleGAN2 as a backbone generative model of images. We denote the image generator as $G : \mathcal{Z} \to \mathcal{W} \to \mathcal{X}$, which is trained following standard StyleGAN training, see for more information [1, 2]. In particular, we use the pre-trained Car, Face-FFHQ and Cat StyleGAN2 models from the official GitHub repository provided by StyleGAN2[2]. For Bird, we use the StyleGAN2 model trained on NABirds-48k [3].

The StyleGAN2 generator maps latent codes $\mathbf{z} \in \mathcal{Z}$, drawn from a multivariate Normal distribution, $\mathcal{N}(\mathbf{z}; \mathbf{0}, \mathbf{I})$, into realistic images. A latent code $\mathbf{z}$ is first transformed into an intermediate code $\mathbf{w} \in \mathcal{W}$ by a non-linear mapping function $m(\mathbf{z})$. $\mathbf{w}$ is then further transformed into $K+1$ independent vectors, $\mathbf{w}^0, ..., \mathbf{w}^K$, through $K+1$ learned affine transformations. These $K+1$ transformed latent codes are fed into synthesis blocks, sometimes called style layers and denoted as $\{\text{Style}^0, \text{Style}^1, .., \text{Style}^K\}$ [4]. The output of these synthesis blocks are deep feature maps $\{S^0, S^1, ..., S^K\}$. These feature maps carry the information for forming the image $\mathbf{x} \in \mathcal{X}$, which is achieved by connecting them to a residual image synthesis branch. Further details and visualizations about the StyleGAN2 architecture can be found in [1, 2].

### 1.2  Image Encoder

To embed images into the GAN's latent space, the EditGAN framework relies on optimization, initialized by an encoder. To train this encoder we mainly follow SemanticGAN [5], which builds on [6], with further improvements.

We start by introducing notation. Let us denote $\mathbb{D}_{\mathbf{x}}$ as a dataset of real images and $\mathbb{D}_{\mathbf{x},\mathbf{y}}$ as a dataset of image-segmentation mask pairs. Note that the number of images in the unannotated data $\mathbb{D}_{\mathbf{x}}$ is usually much larger than the annotated $\mathbb{D}_{\mathbf{x},\mathbf{y}}$. In fact, $\mathbb{D}_{\mathbf{x},\mathbf{y}}$ is as small as 16 or 30 image-segmentation pairs for our datasets. We directly embed the images into $\mathcal{W}^+$ space, where the $K+1$ $\mathbf{w}^0, ..., \mathbf{w}^K$ are modeled independently for each style layer [7]. Thus, we can formally define a variation of the generator as $\hat{G} : \mathcal{W}^+ \to \mathcal{X}$, which operates on this $\mathcal{W}^+$ space. We follow [6] and train an encoder

---

[*]These authors contributed equally.

[2]https://github.com/NVlabs/stylegan2 (Nvidia Source Code License)

35th Conference on Neural Information Processing Systems (NeurIPS 2021), Sydney, Australia.

$E_\phi : \mathcal{X} \to \mathcal{W}^+$ with parameters $\phi$ using the following objective functions:

$$\mathcal{L}_{\text{RGB}}(\phi) = \mathbb{E}_{\mathbf{x} \in \mathbb{D}_\mathbf{x}} \left[ \lambda_1 L_{\text{LPIPS}}(\mathbf{x}, \ \hat{G}(E_\phi(\mathbf{x}))) + \lambda_2 L_{\text{L2}}(\mathbf{x}, \ \hat{G}(E_\phi(\mathbf{x}))) \right] \tag{1}$$

where $L_{\text{LPIPS}}$ loss is the Learned Perceptual Image Patch Similarity (LPIPS) distance [8] and $L_{\text{L2}}$ is a standard L2 loss. We also explicitly regularize the encoder output distribution using an additional loss that utilizes samples from the GAN itself:

$$\mathcal{L}_{\text{Sampling}}(\phi) = \mathbb{E}_{\mathbf{x} = G(\mathbf{z}), \mathbf{z} \sim \mathcal{N}(\mathbf{z}; \mathbf{0}, \mathbf{I})}[\lambda_3 L_{\text{LPIPS}}(\mathbf{x}, \ \hat{G}(E_\phi(\mathbf{x}))) \tag{2}$$

$$+ \lambda_4 L_{\text{L2}}(\mathbf{x}, \ \hat{G}(E_\phi(\mathbf{x}))) + \lambda_5 L_{\text{L2}}(m(\mathbf{z}), \ E_\phi(\mathbf{x}))] \tag{3}$$

Here, $m(\mathbf{z})$ is the previously introduced mapping function $m : \mathcal{Z} \to \mathcal{W}$ and $\lambda_{1,\dots,5}$ are hyperparameters. For all classes, we set $\lambda_1 = 10$, $\lambda_2 = 1$, $\lambda_3 = 10$, $\lambda_4 = 1$, and $\lambda_5 = 5$. We use the Adam [9] optimizer with learning rate $3 \times 10^{-5}$ and batch size 8 to train the encoder. Experimentally, for the Car and Cat datasets, we first train only on samples from the GAN itself using Eq. 3 for 20,000 iterations as warm up, and then train jointly using Eq. 1 and Eq. 3 iteratively until the model converges on the training dataset.

After successful encoder training, to embed images we first use the encoder $E_\phi$ and further iteratively refine the latent code $\mathbf{w}^+$ via optimization with respect to the $\mathcal{L}_{\text{RGB}}$ objective (without further modifying encoder parameters $\phi$). We run optimization for 500 steps with $\lambda_1 = 10$, $\lambda_2 = 1$. We use the Adam [9] optimizer with the lookahead technique [10] with a constant learning rate of 0.001.

## 1.3 Segmentation Branch

Using our encoder together with additional optimization, as described in the previous section, we embed the annotated images $\mathbf{x}$ from $\mathbb{D}_{\mathbf{x}, \mathbf{y}}$ into $\mathcal{W}^+$, formally constructing $\mathbb{D}_{\mathbf{x}, \mathbf{y}, \mathbf{w}^+}$, the annotated dataset augmented with $\mathbf{w}^+$ embeddings.

Similar to DatasetGAN [11], to generate segmentation maps $\mathbf{y}$ alongside images $\mathbf{x}$ we then train a segmentation branch $I_\psi$ with parameters $\psi$. $I_\psi$ is a simple three-layer multi-layer perceptron classifier on the layer-wise concatenated and appropriately upsampled feature maps. Specifically, the lower-resolution deep feature maps in $\{S^0, S^1, ..., S^K\}$ are first appropriately upsampled, $\hat{S}^k = U_k(S^k)$ for $k \in 0, ..., K$ and upsampling functions $U_k$, so that all feature maps have the same spatial resolution, equal to the highest resolution, and can be concatenated channel-wise. The classifier operates on the layer-wise concatenated feature maps in a per-pixel fashion and predicts the segmentation label of each pixel. It is trained via the objective

$$\mathcal{L}_I(\psi) = \mathbb{E}_{\mathbf{x}, \mathbf{y}, \mathbf{w}^+ \in \mathbb{D}_{\mathbf{x}, \mathbf{y}, \mathbf{w}^+}} \left[ H(\mathbf{y}, I_\psi((\hat{S}^0, \hat{S}^1, ..., \hat{S}^K))) \right], \tag{4}$$

$$\text{with} \quad \hat{S}^k = U_k(\text{Style}_k(\mathbf{w}_k^+)), \tag{5}$$

where $I_\psi$ takes as input the concatenated and appropriately upsampled feature maps $(\hat{S}^0, \hat{S}^1, ..., \hat{S}^K)$. We use bilinear-upsampling operations $U_k$. Furthermore, $H$ denotes the pixel-wise cross-entropy.

To train the segmentation branch $I_\psi$ and minimize the objective $\mathcal{L}_I(\psi)$, we use the Adam optimizer with learning rate 0.001. We randomly sample 64 pixels across all training images for each batch. The segmentation branch is trained until it converges on the training dataset. After training the segmentation branch $I_\psi$ we can formally define a generator $\tilde{G} : \mathcal{W}^+ \to \mathcal{X}, \mathcal{Y}$ that models the joint distribution $p(\mathbf{x}, \mathbf{y})$ of images $\mathbf{x}$ and semantic segmentations $\mathbf{y}$.

Notice that we defined the segmentation branch here using a new symbol, $I_\psi$, opposed to $\tilde{G}^\mathbf{y}$ from the main paper. Here, $I_\psi$ specifies the specific network that is only part of the segmentation branch and acts on top of the feature maps $(\hat{S}^0, \hat{S}^1, ..., \hat{S}^K)$ in a pixel-wise manner. On the other hand, the segmentation generation component $\tilde{G}^\mathbf{y}$, defined in the main paper, denotes the complete segmentation generation module, starting from $\mathcal{W}^+$, including the style layers and deep feature maps that are shared between the image and segmentation generation branches.

## 1.4 Learning Editing Vectors

To perform editing and learn editing vectors, we proceed as described in detail in Secs. 3.3 and 3.4 in the main text. The ArcFace feature extraction network checkpoint [12] is taken from `https://github.com/TreB1eN/InsightFace_Pytorch` (MIT License).

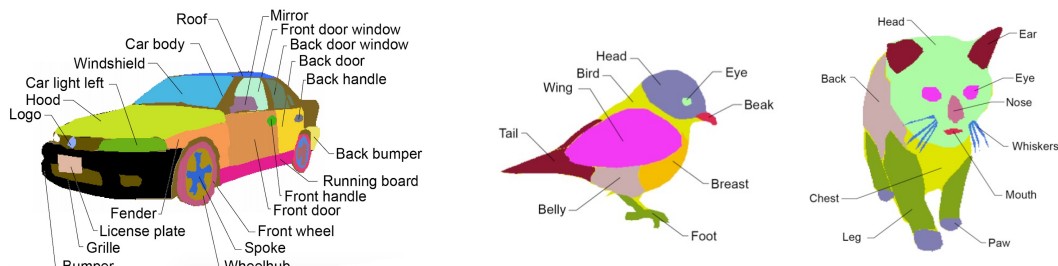

Figure 1: Car, Bird and Cat part labeling schemes [11].

In the main paper, we already provided the label scheme for the Face data (Fig. 6). The further labeling schemes for the Car, Bird, and Cat classes are shown in Fig 1. The annotations contain 34, 32, 16, and 11 possible pixel labels for the Face, Car, Bird and Cat data, respectively. When performing optimization to find the editing vectors $\delta\mathbf{w}_{\text{edit}}^+$, we use the Adam [9] optimizer with learning rate 0.02 and run for 100 steps. We use the hyperparameters $\lambda_1^{\text{editing}} = 15$, $\lambda_2^{\text{editing}} = 1$, and $\lambda_3^{\text{editing}} = 10$ (Eq. 5 in main paper). When performing optimization for self-supervised refinement after initializing the edit with an editing vector (as described in second bullet point in Sec. 3.4 in main paper), we use the same optimizer and we set hyperparameters $\lambda_1^{\text{editing}} = 5$, $\lambda_2^{\text{editing}} = 1$, and $\lambda_3^{\text{editing}} = 5$. Hyperparameters are chosen based on visual quality on hold-out examples. We will release the training set $\mathbb{D}_{\mathbf{x},\mathbf{y}}$ and learnt editing vectors.

## 2 Experiment Details

Here, we provide additional experiment details.

### 2.1 Smile Edit Benchmark

In Section **4.2** of the main paper, we evaluate our model against strong baselines on the smile edit benchmark introduced by MaskGAN [13]. Here we provide more details for completeness. **Semantic Correctness:** To measure whether the faces show smiling expressions after editing, a binary smile attribute classifiers is trained on the CelebA training set, using a ResNet-18 [14] backbone. The input faces are resized into resolution of $256 \times 256$. The classifier achieives 92.2% accuracy on the CelebA testing dataset. **Identity Preservation:** We again use the pretrained ArcFace feature extraction network [12] with checkpoint from https://github.com/TreB1eN/InsightFace_Pytorch (MIT License). As pointed out in main paper, we did not use the identity loss in this benchmark experiment when performing face editing. In this benchmark experiment, this facial feature extraction network is used *only* for evaluation purposes.

To compare with the baselines, we took the officially released MaskGAN[3] [13] and LocalEditing[4] [15] checkpoints. Furthermore, we train an InterFaceGAN [16] smile model using the officially released code[5] where we replaced the generator with a StyleGAN2 for fair comparison. At inference time when performing editing, we use the same test image embeddings for the InterFaceGAN model as we use for our EditGAN model. As mentioned in the main paper, we also use StyleGAN2 Distillation [17] as baseline, for which we rely on the official codebase[6] and train a Pix2PixHD network for the smile edit using the default hyperparameters as provided in the paper [17]. We further show in Fig. 2 the image and initial and modified segmentation masks that were used to learn our smile editing vector.

Finally, we provide more details for Fig. 10 in the main text: For each curve, we report results with five different editing vector scale coefficients $s_{\text{edit}} \in [0.7, 1, 1.3, 1.5, 1.7]$.

---

[3] https://github.com/switchablenorms/CelebAMask-HQ
[4] https://github.com/IVRL/GANLocalEditing
[5] https://github.com/genforce/interfacegan
[6] https://github.com/EvgenyKashin/stylegan2-distillation

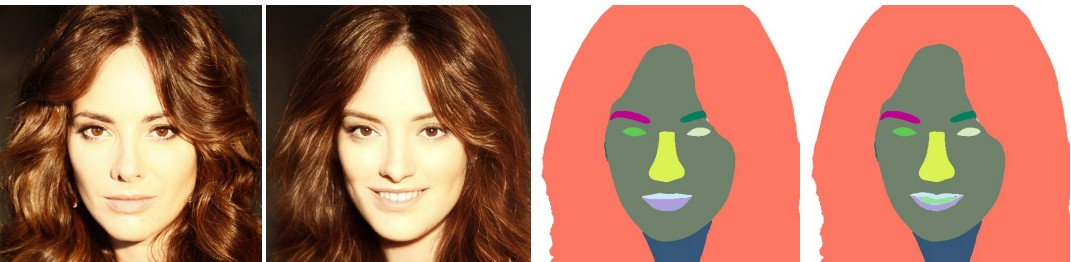

Figure 2: **Image and mask pair to learn smile editing vector on CelebA.** Images are face before editing, face after editing, segmentation mask predicted by segmentation branch before editing (after embedding the image into EditGAN's latent space), and target segmentation mask after manual modification.

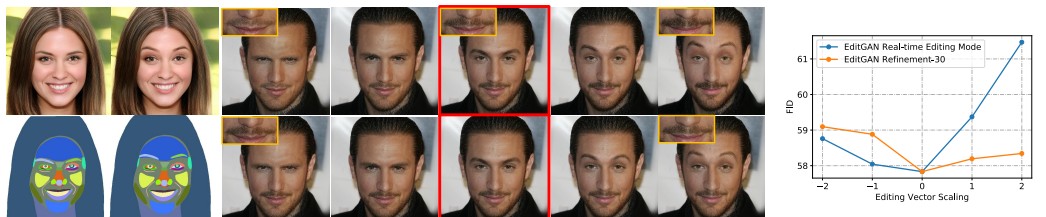

Figure 3: *Left:* We apply learnt editing vectors with varying scales (see 5 markers in FID plots) both without (top row) and with (bottom row) additional 30-step self-supervised refinement to correct artifacts. Red boxes denote original images. For each class, the leftmost image is the one used to learn the editing vector with the editing result next to it and orginal and modified segmentations below. *Right:* Visual quality after editing with different scales as measured by FID with and without 30-step refinement.

## 2.2 Additional Results: Editing Vector Scale Experiment

In the main paper, in Fig. 9, we presented another ablation study where we studied editing quality when applying edits with different editing vector scales $s_{\text{edit}}$. We analyzed editing quality both visually and quantitatively, both with and without self-supervised refinement. While in the main paper we only presented the results for Car and Cat data, here we additionally show the results on Face images, using an edit that raises eyebrows as example (Fig. 3). Similar to the other results on Car and Cat data, we find that editing by purely applying our learnt editing vector, which can be done at interactive rates, already yields virtually perfect editing results. However, we do observe almost unnoticeable entanglement with the beard. Using self-supervised refinement we can fully remove this editing artifact, if necessary.

## 2.3 Additional Results: Smile Edit Benchmark with more Test Images

In Tab. 1 of the main paper, we use MaskGAN's [13] smile edit benchmark. The FID scores are calculated between 400 edited test images and the CelebA-HD test database, which enables a fair comparison with existing approaches and directly follows the practice by MaskGAN. Although the estimates may be biased with respect to the true FID due to the limited number of test images [18], we expect that they nevertheless provide a fair comparison between

| Metric | # Mask Annot. | # Attribute Annot. | Attribute Acc.(%) ↑ | FID ↓ | ID Score ↑ |
|---|---|---|---|---|---|
| MaskGAN [13] | 30,000 | - | 65.7 | **18.3** | 0.5229 |
| LocalEditing [15] | - | - | 23.7 | 20.4 | 0.5726 |
| InterFaceGAN [16] | - | 30,000 | 79.5 | 34.4 | 0.6560 |
| EditGAN (ours) | 16 | - | **88.0** | 32.1 | 0.6422 |
| EditGAN$^{+}$30 (ours) | 16 | - | 81.4 | 31.8 | **0.6625** |

Table 1: Quantitative comparisons to multiple baselines on the smile edit 4k benchmark.

the different methods. However, here we re-calculate FID as well as attribute accuracy and ID score using 10 times as many images, i.e. 4000 images, from the training set from MaskGAN. Notice that only MaskGAN uses this data for training, while the GANs of all other baselines, including our EditGAN, are based on the FFHQ faces data and do not use this annotated training data that MaskGAN relies on. Hence, calculating the FID using these 4000 images is advantageous for MaskGAN.

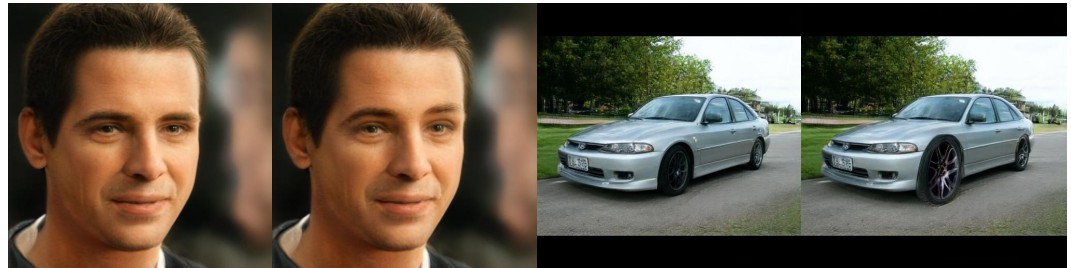

Figure 4: We demonstrate challenging editing operations where we disentangle semantically related parts. The presented results correspond to pure optimization-based editing. *First example*: Lift right eyebrow while keeping the left eyebrow unchanged. *Second example*: Enlarge the front wheel while keeping the back wheel unchanged.

We show results in Tab. 1. Since we use different and much more data for evaluation compared to the evaluation reported in the table in the paper, the numbers are different. However, the rankings and comparisons between the methods remain the same and the conclusions are the same. In particular, EditGAN achieves the best attribute accuracies and ID scores. MaskGAN achieves a relatively low FID, but this is simply due to the unfair comparison, as discussed above. MaskGAN still performs significantly worse than InterfaceGAN and EditGAN in attribute accuracy and ID score.

## 3 Computational Resources

Training of the underlying StyleGAN2, the encoder, and the segmentation branch, as well as optimization for embedding and editing were performed using NVIDIA Tesla V100 GPUs on an in-house GPU cluster. Overall, the project used approximately 14,000 GPU hours (according to internal GPU usage reports), of which around 3,500 GPU hours were used for the final experiments, and the rest for exploration and testing during the earlier stages of the research project.

## 4 Additional Qualitative Results

Below, we present further qualitative results.

We first demonstrate particularly challenging editing operations where we try to disentangle semantically related parts. For example, we want to lift the right eyebrow while keeping the left eyebrow unchanged. We present the results ins Fig. 4. Furthermore, we again demonstrate the ability to combine multiple different edits in Fig 5. We also invite the reader to watch our video, which shows latent code interpolations between the edits. Finally, for all edits we perform in the main paper, we first show the image and segmentation mask pairs that were used to learn the latent space editing vectors, and then we present a few more editing results on GAN-generated images (Figs. 6-22).

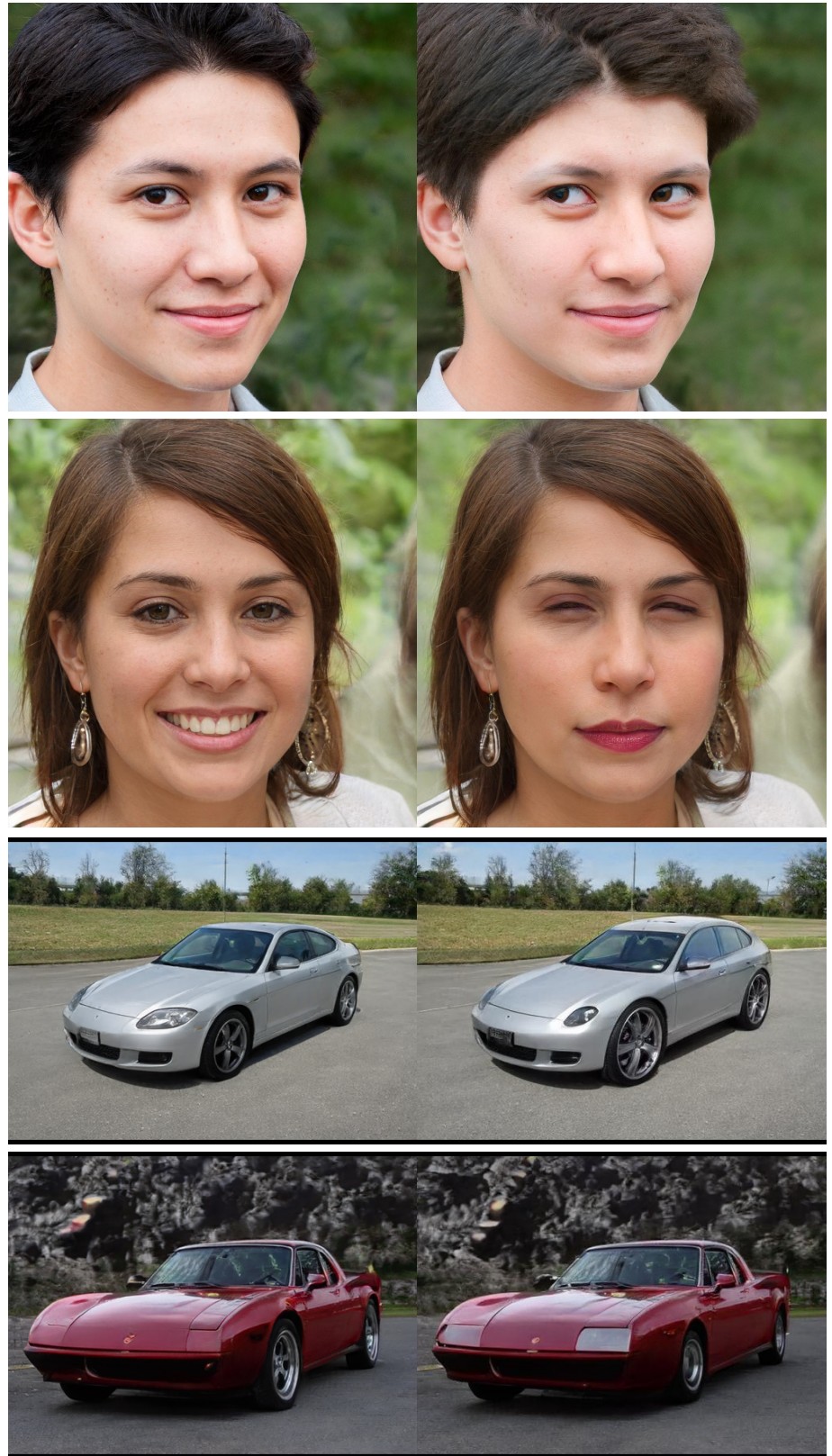

Figure 5: **Combining multiple edits.** Results are based on editing with learnt editing vector and 30 steps of self-supervised refinement. Edits in detail: *First row*: Slight frown, look left, add hair, remove smile wrinkle. *Second row*: Close eyes, close mouth, remove smile wrinkle. *Third row*: Lift back of the car, enlarge wheels, shrink front light. *Fourth row*: Enlarge front light, shrink wheels. Please also see attached video which shows latent code interpolations between editing operations.

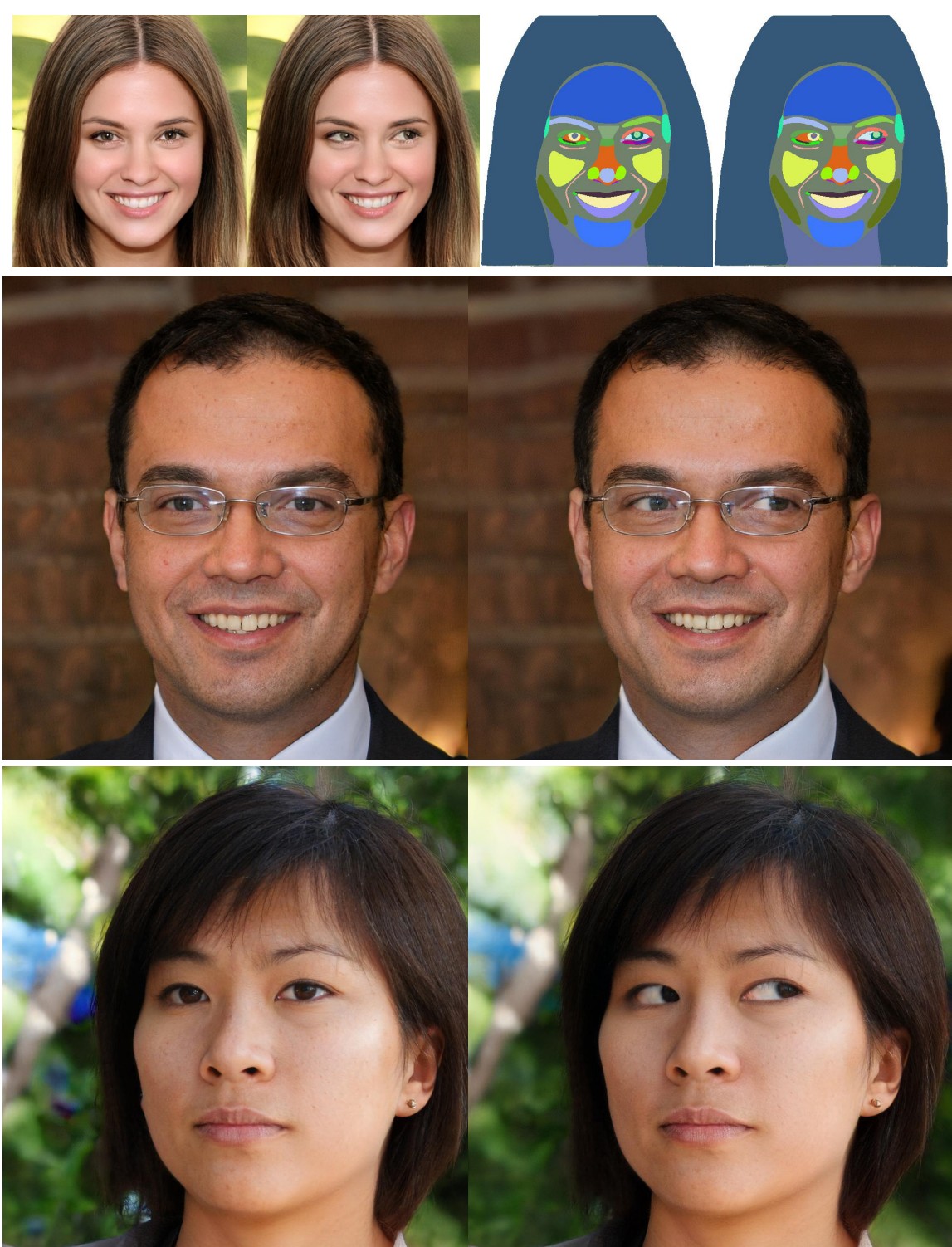

Figure 6: **Gaze position editing.** *First row*: Image and mask pair to learn editing vector. Images are images before editing and after editing. Segmentation masks are before editing and target segmentation mask after manual modification. *Second and third rows*: Applying the learnt edit on new images.

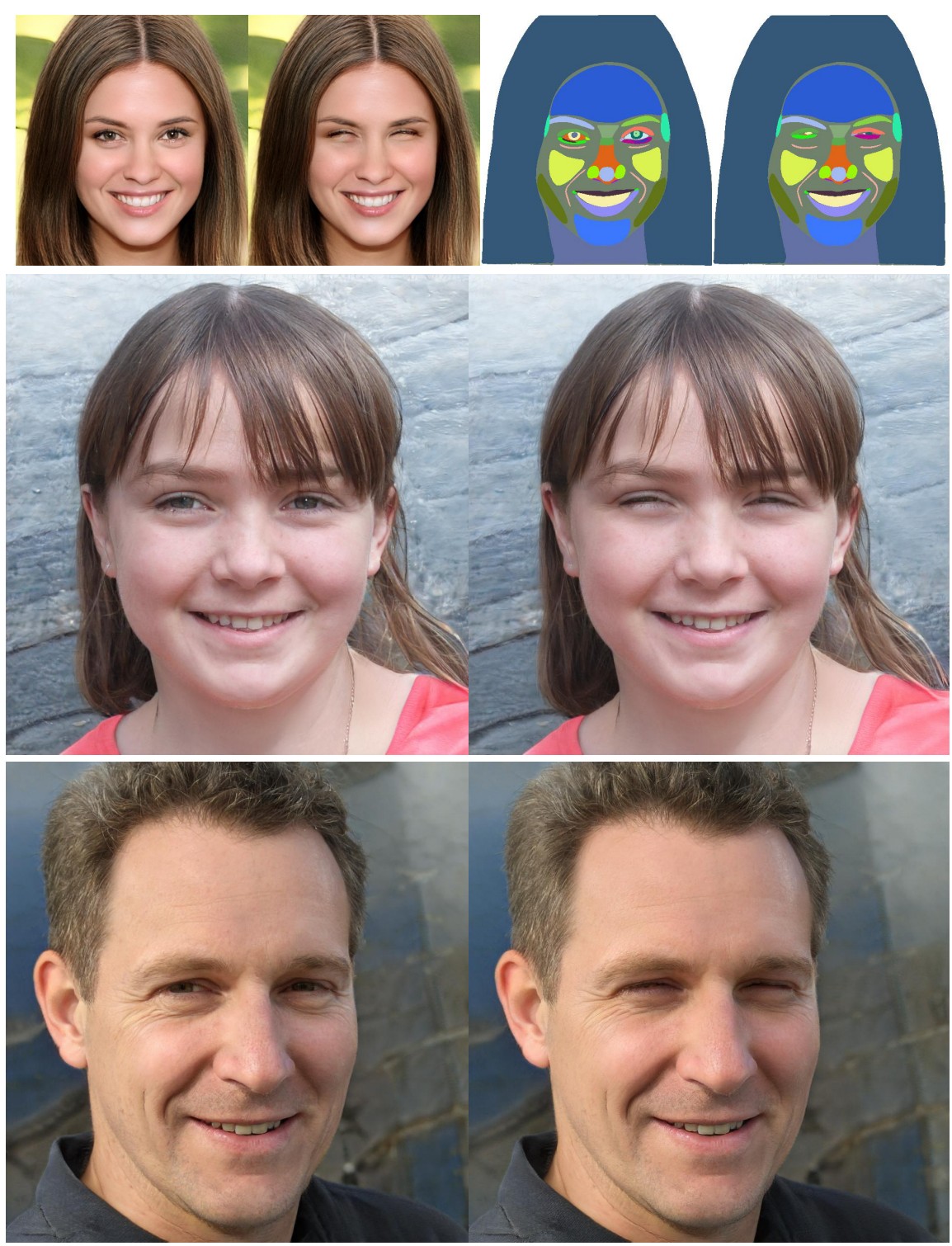

Figure 7: **Closing eyes editing.** *First row*: Image and mask pair to learn editing vector. Images are images before editing and after editing. Segmentation masks are before editing and target segmentation mask after manual modification. *Second and third rows*: Applying the learnt edit on new images.

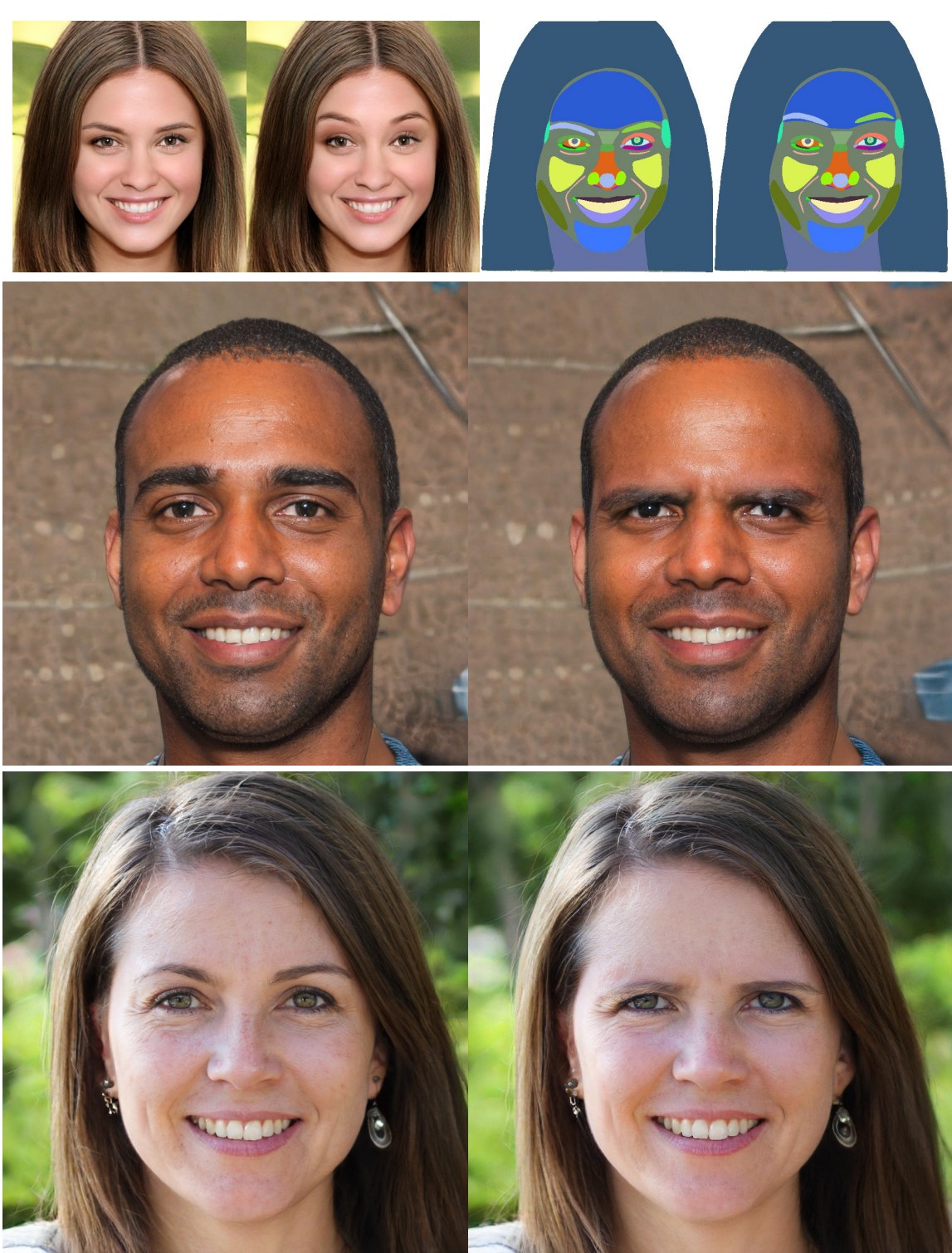

Figure 8: **Raising eyebrows editing.** *First row*: Image and mask pair to learn editing vector. Images are images before editing and after editing. Segmentation masks are before editing and target segmentation mask after manual modification. *Second and third rows*: Applying the learnt edit on new images (with flipped direction, i.e. negative editing vector scale $s_{\text{edit}}$.).

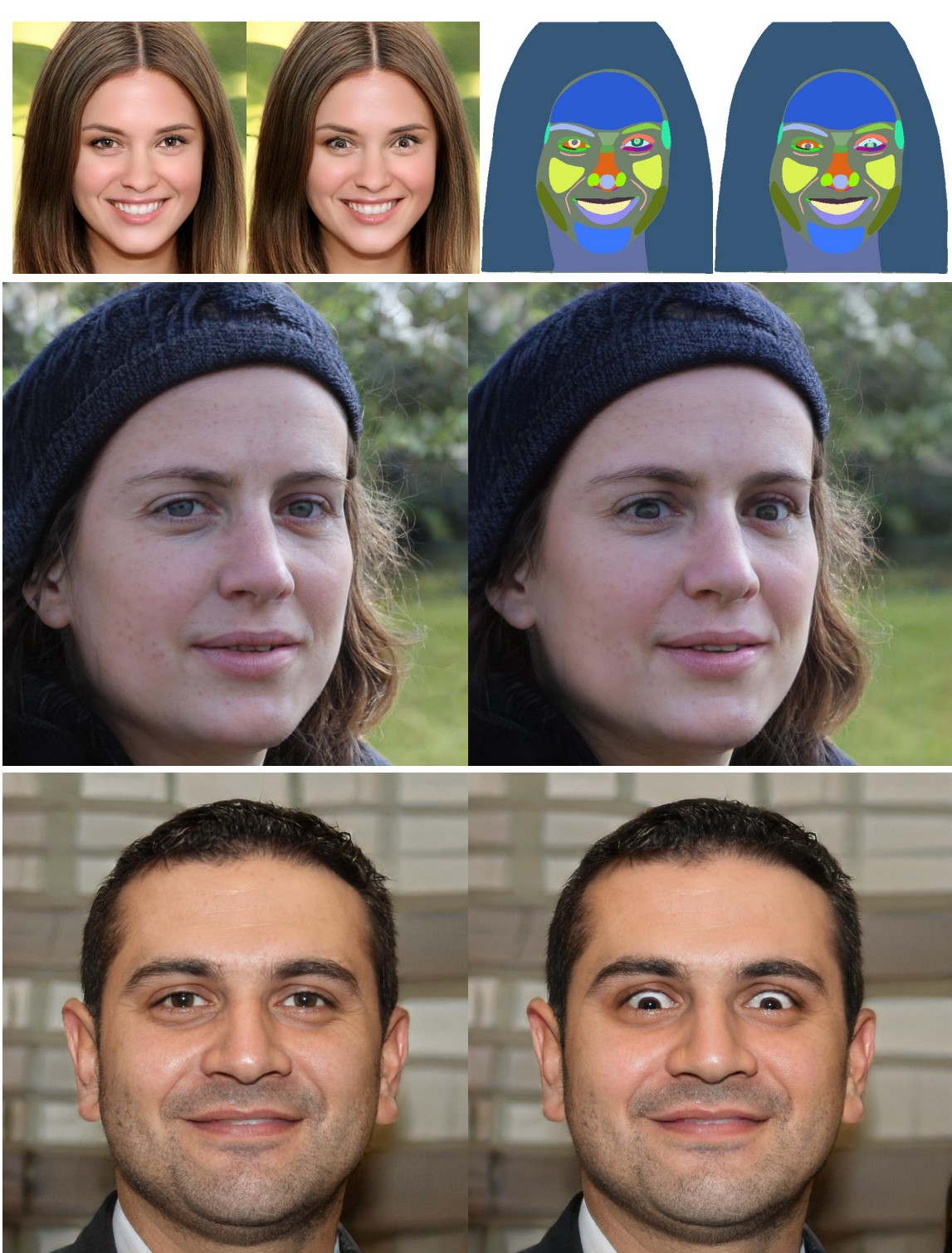

Figure 9: **Vertical gaze position editing.** *First row*: Image and mask pair to learn editing vector. Images are images before editing and after editing. Segmentation masks are before editing and target segmentation mask after manual modification. *Second and third rows*: Applying the learnt edit on new images.

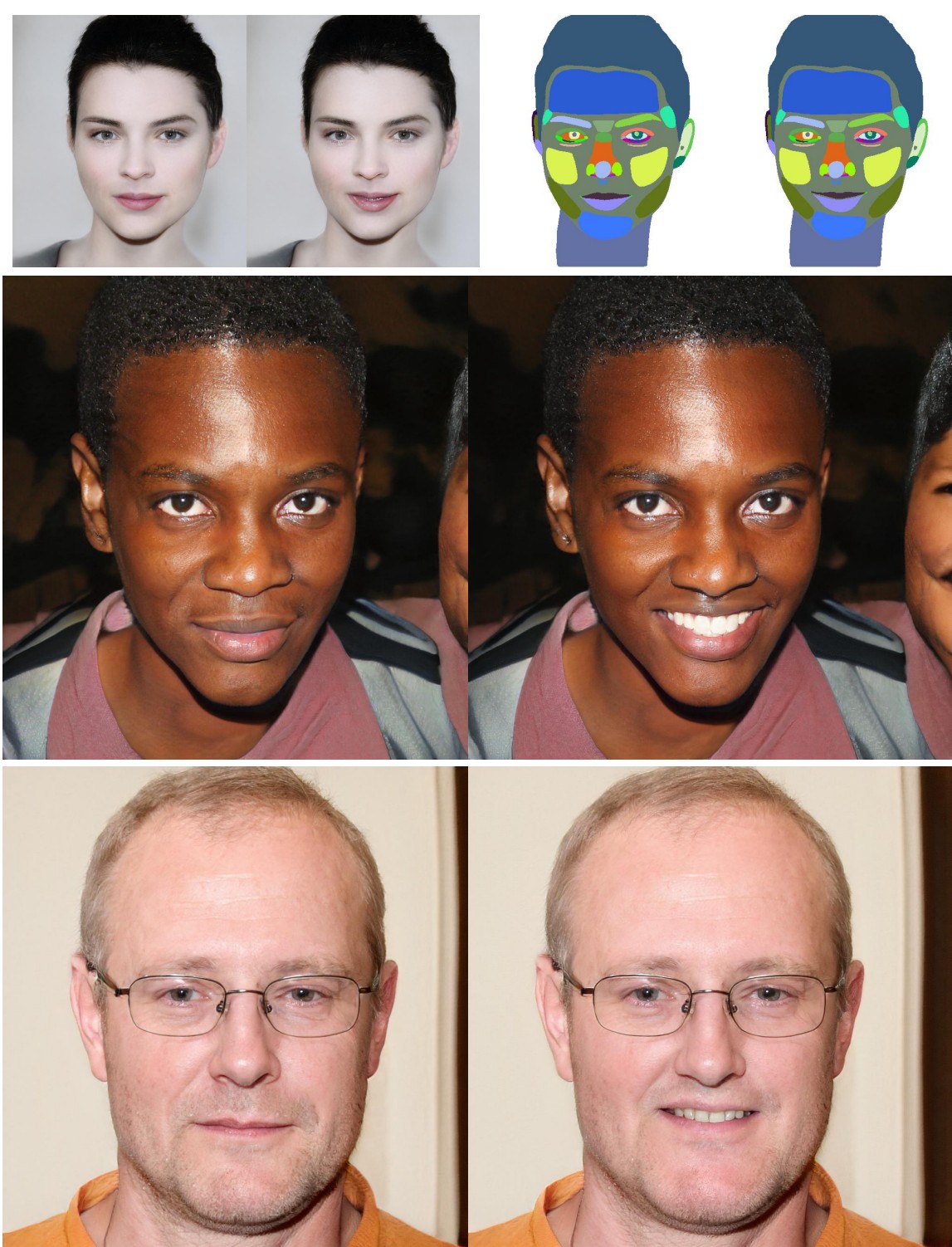

Figure 10: **Smile editing.** *First row*: Image and mask pair to learn editing vector. Images are images before editing and after editing. Segmentation masks are before editing and target segmentation mask after manual modification. *Second and third rows*: Applying the learnt edit on new images.

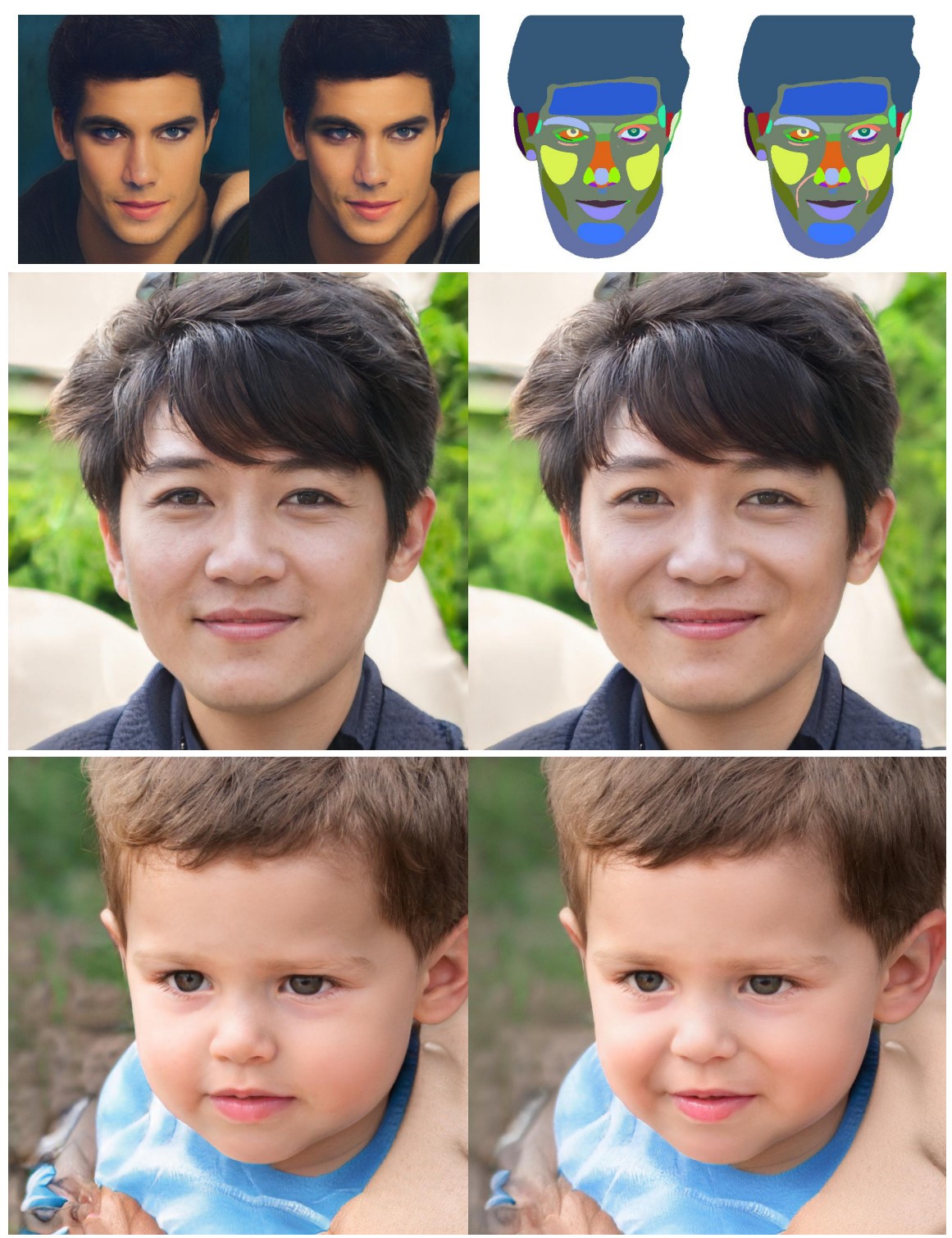

Figure 11: **Adding smile wrinkle editing.** *First row*: Image and mask pair to learn editing vector. Images are images before editing and after editing. Segmentation masks are before editing and target segmentation mask after manual modification. *Second and third rows*: Applying the learnt edit on new images.

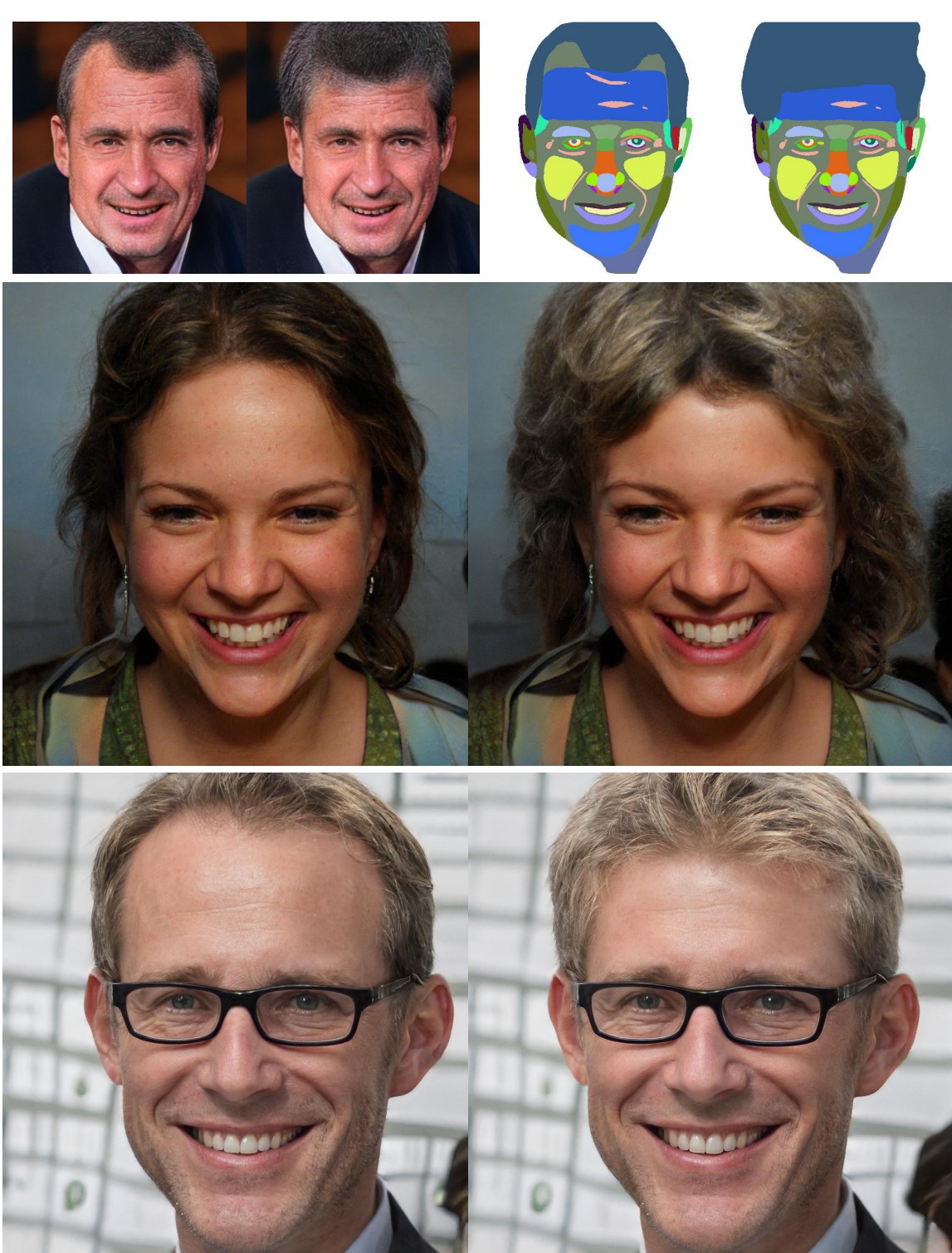

Figure 12: **Hairstyle editing.** *First row*: Image and mask pair to learn editing vector. Images are images before editing and after editing. Segmentation masks are before editing and target segmentation mask after manual modification. *Second and third rows*: Applying the learnt edit on new images.

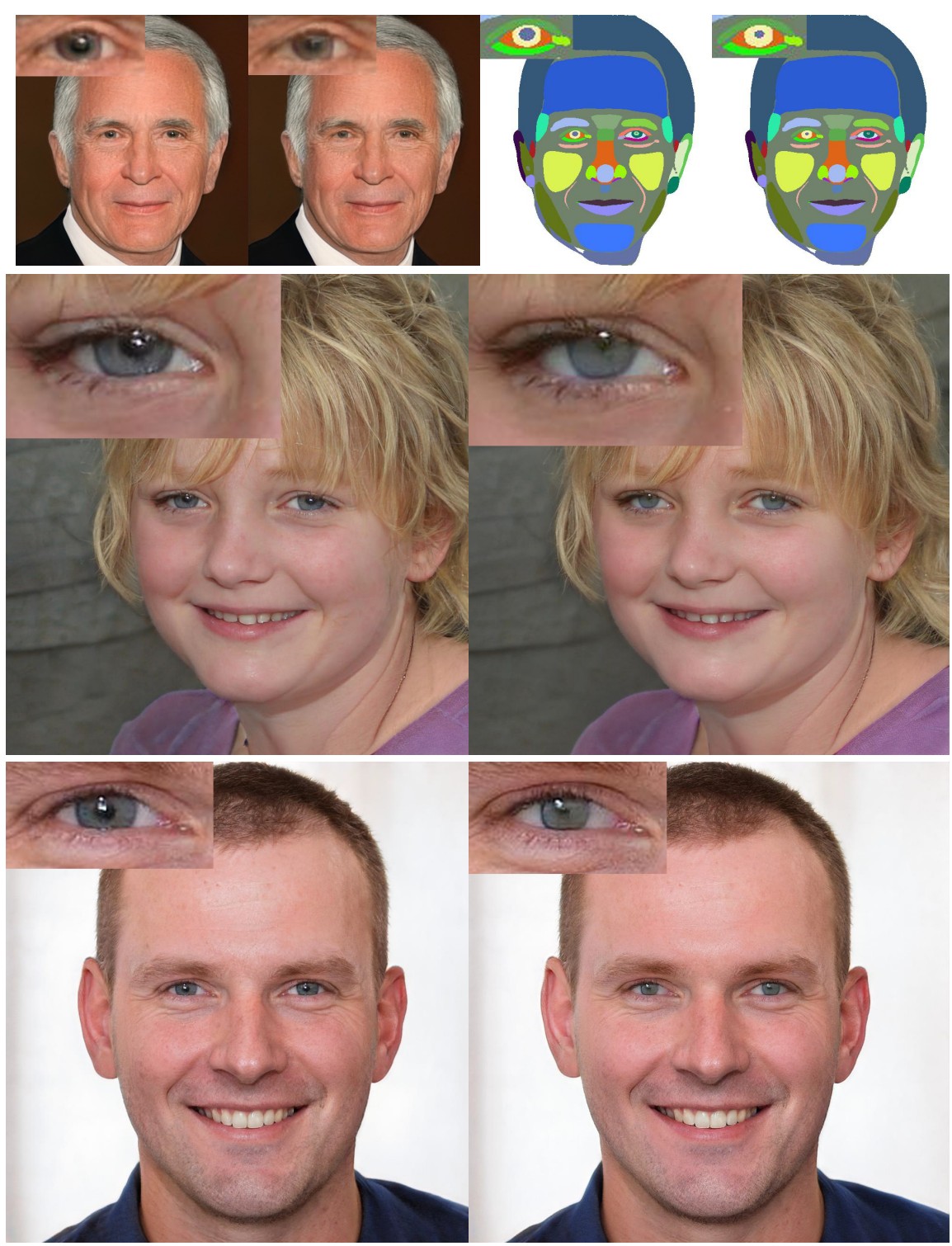

Figure 13: **Pupil size editing.** *First row*: Image and mask pair to learn editing vector. Images are images before editing and after editing. Segmentation masks are before editing and target segmentation mask after manual modification. *Second and third rows*: Applying the learnt edit on new images.

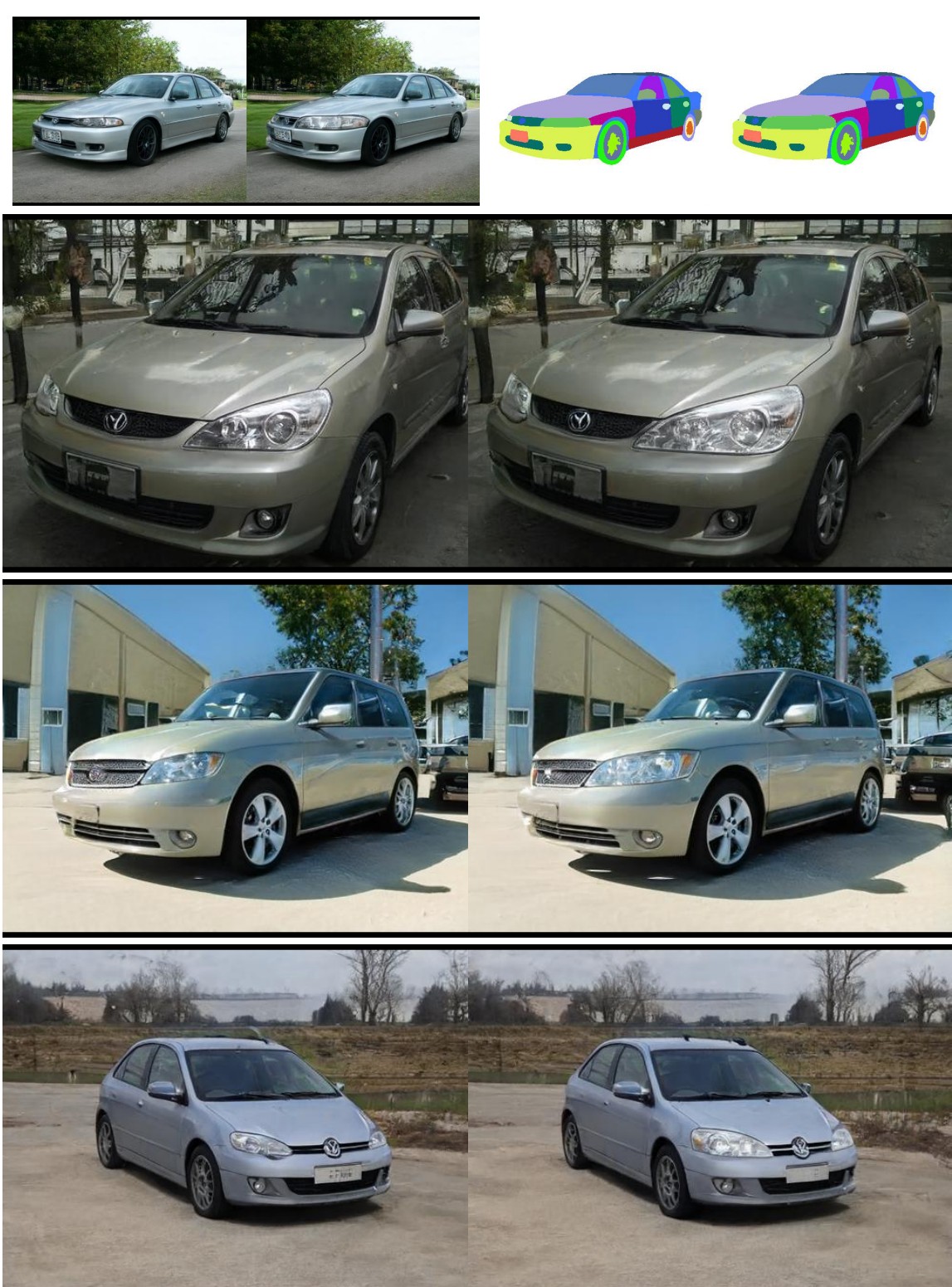

Figure 14: **Front light size editing.** *First row*: Image and mask pair to learn editing vector. Images are images before editing and after editing. Segmentation masks are before editing and target segmentation mask after manual modification. *Second to fourth rows*: Applying the learnt edit on new images.

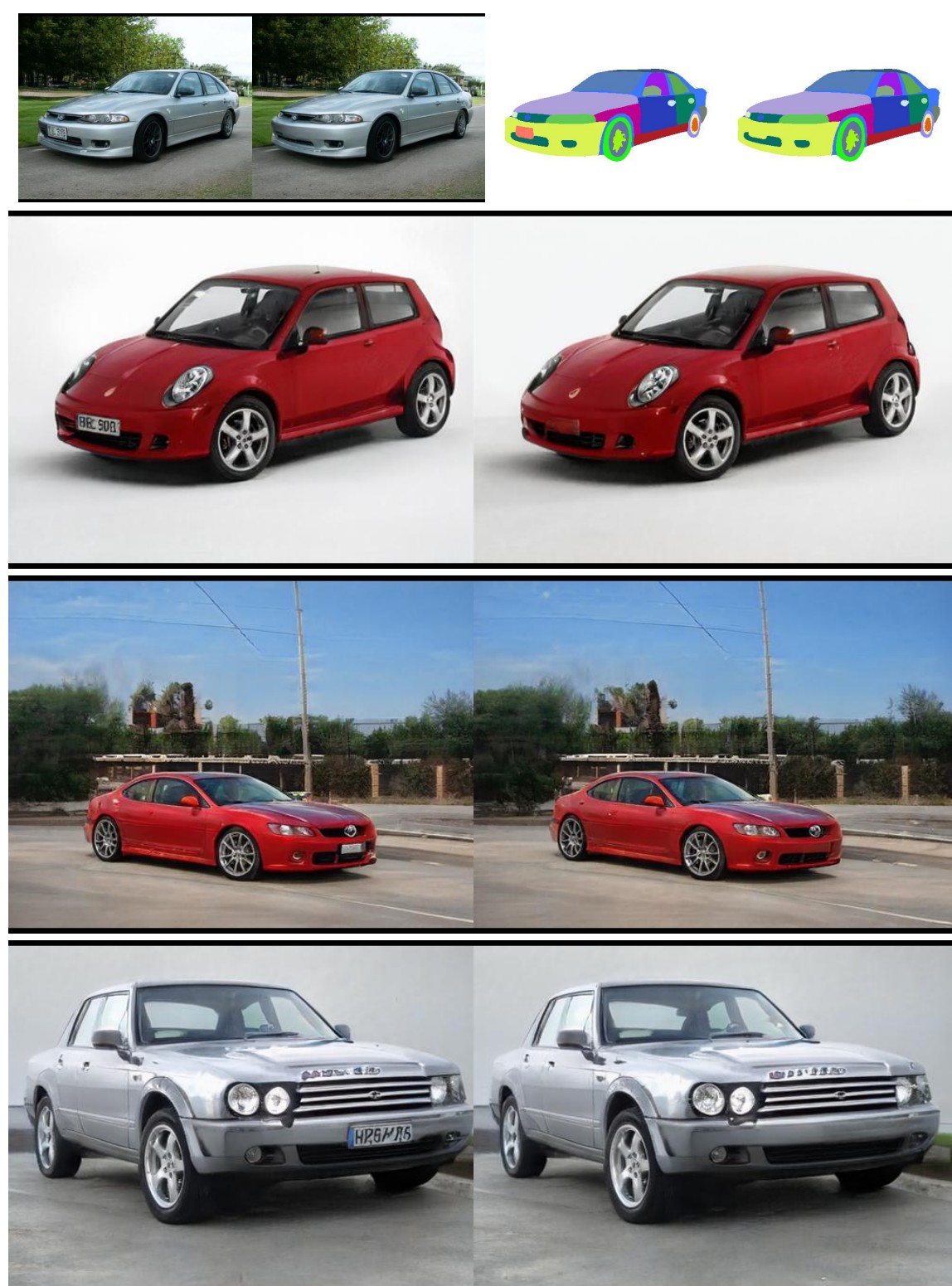

Figure 15: **License plate deletion editing.** *First row*: Image and mask pair to learn editing vector. Images are images before editing and after editing. Segmentation masks are before editing and target segmentation mask after manual modification. *Second to fourth rows*: Applying the learnt edit on new images.

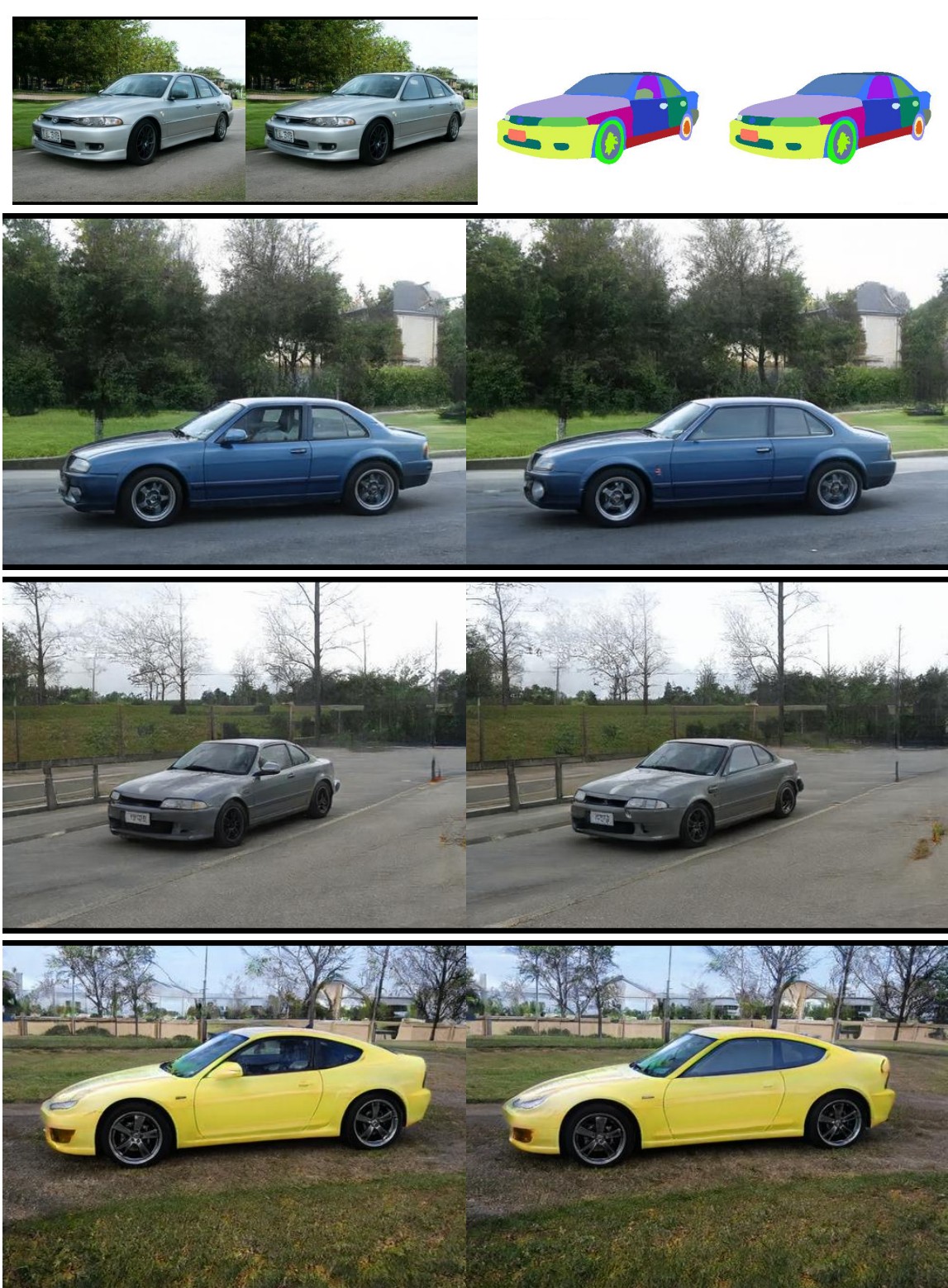

Figure 16: **Side mirror deletion editing.** *First row*: Image and mask pair to learn editing vector. Images are images before editing and after editing. Segmentation masks are before editing and target segmentation mask after manual modification. *Second to fourth rows*: Applying the learnt edit on new images.

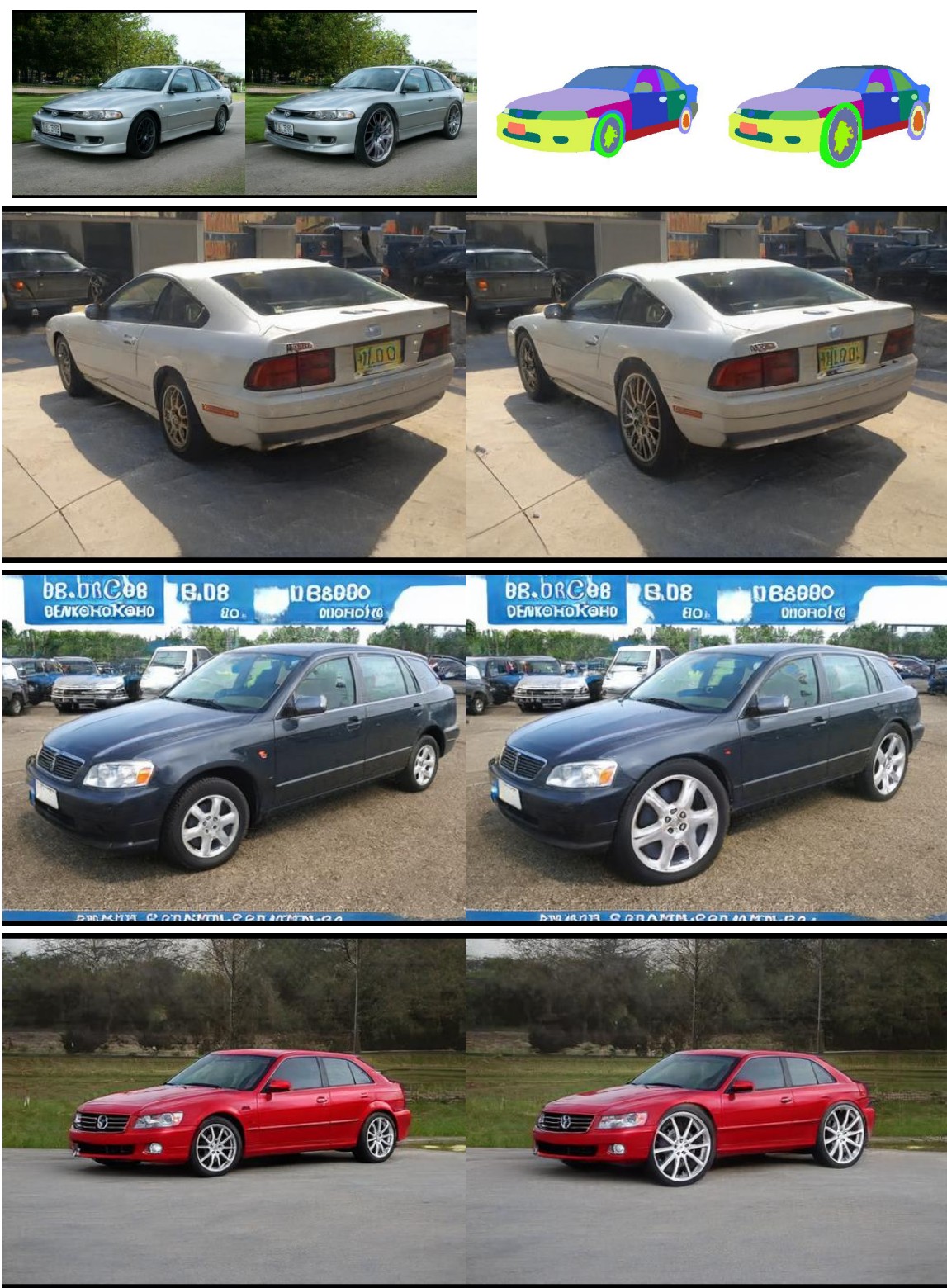

Figure 17: **Wheel size editing.** *First row*: Image and mask pair to learn editing vector. Images are images before editing and after editing. Segmentation masks are before editing and target segmentation mask after manual modification. *Second to fourth rows*: Applying the learnt edit on new images.

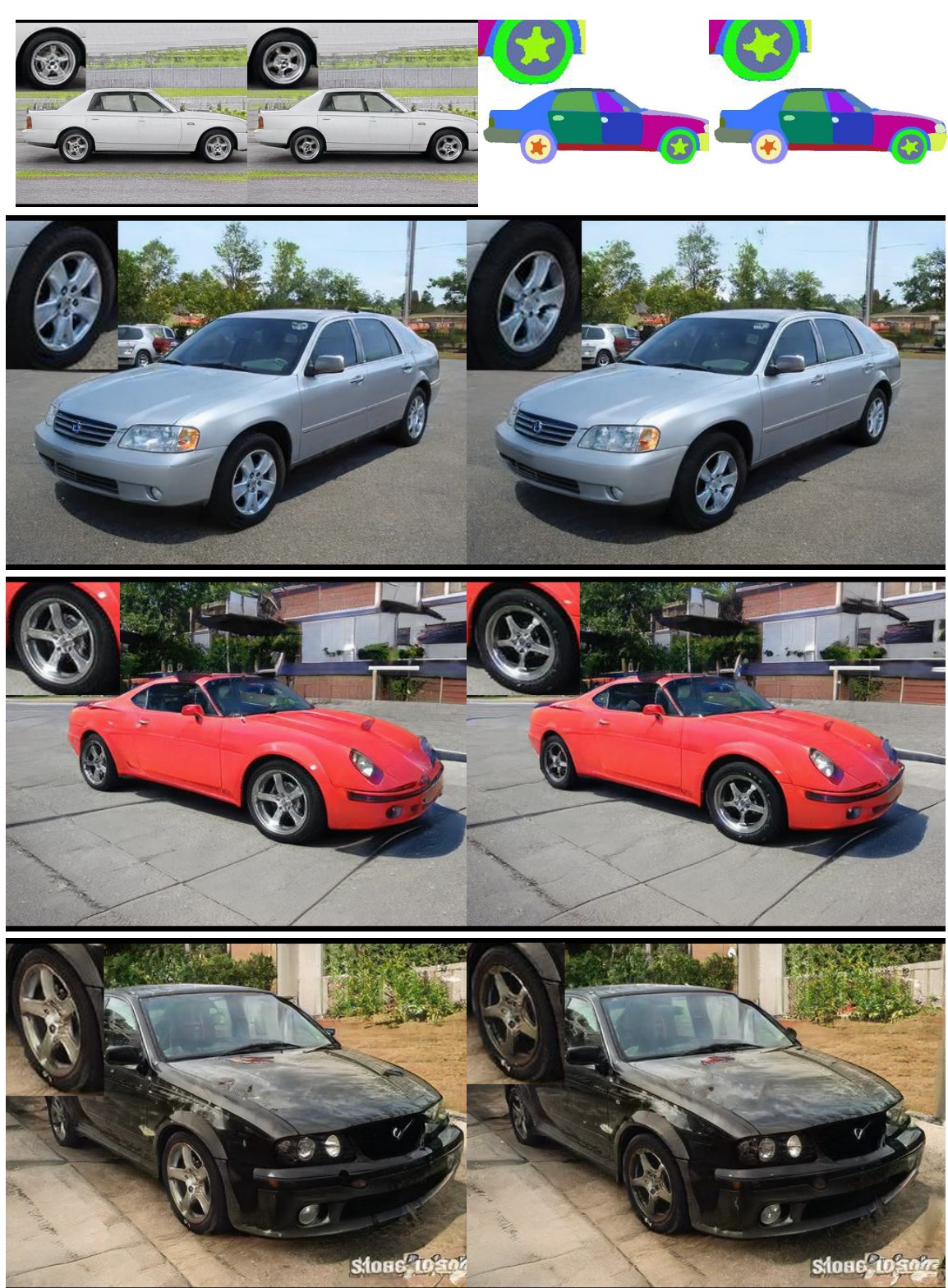

Figure 18: **Wheel/spoke rotation editing.** *First row*: Image and mask pair to learn editing vector. Images are images before editing and after editing. Segmentation masks are before editing and target segmentation mask after manual modification. *Second to fourth rows*: Applying the learnt edit on new images.

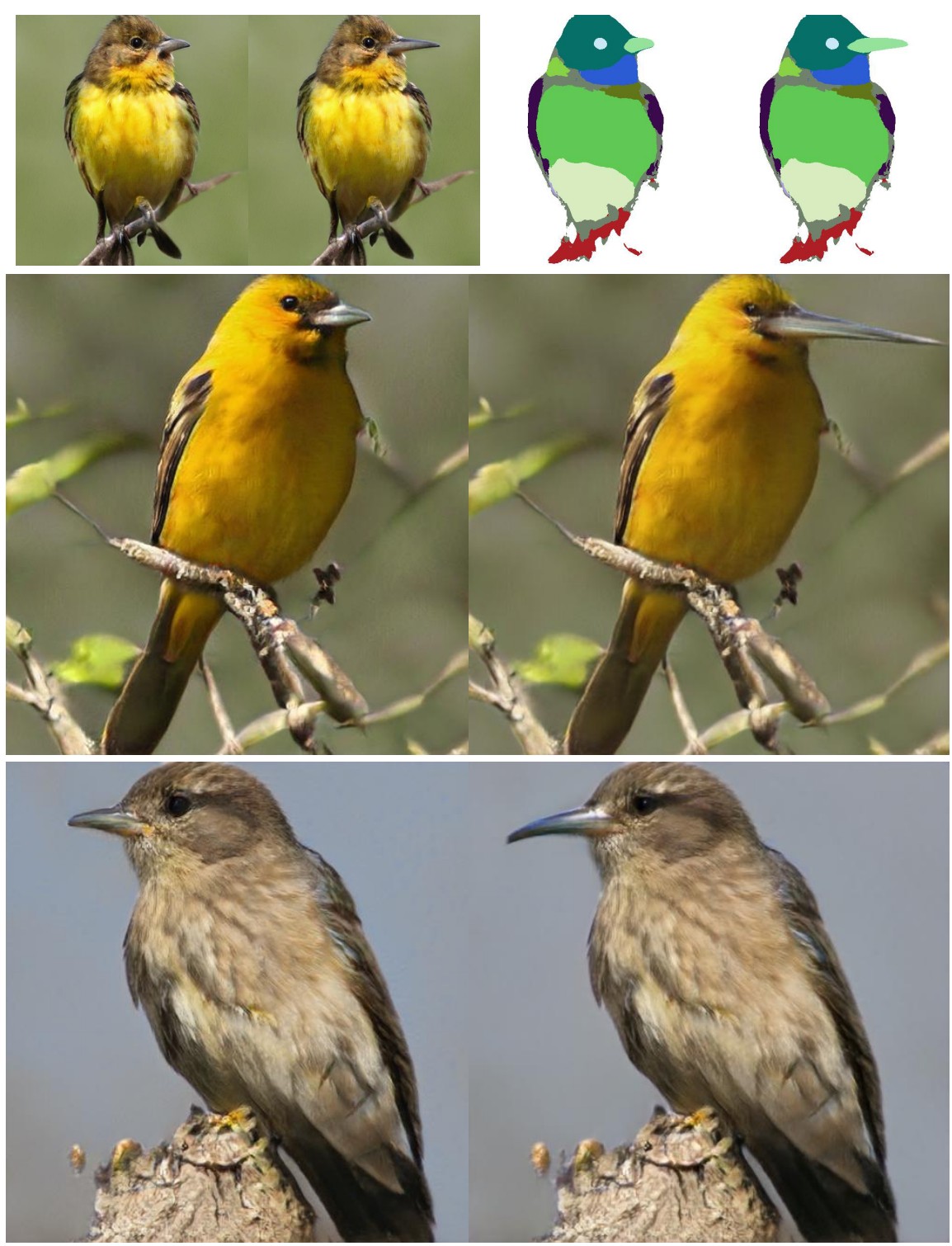

Figure 19: **Beak size editing.** *First row*: Image and mask pair to learn editing vector. Images are images before editing and after editing. Segmentation masks are before editing and target segmentation mask after manual modification. *Second and third rows*: Applying the learnt edit on new images.

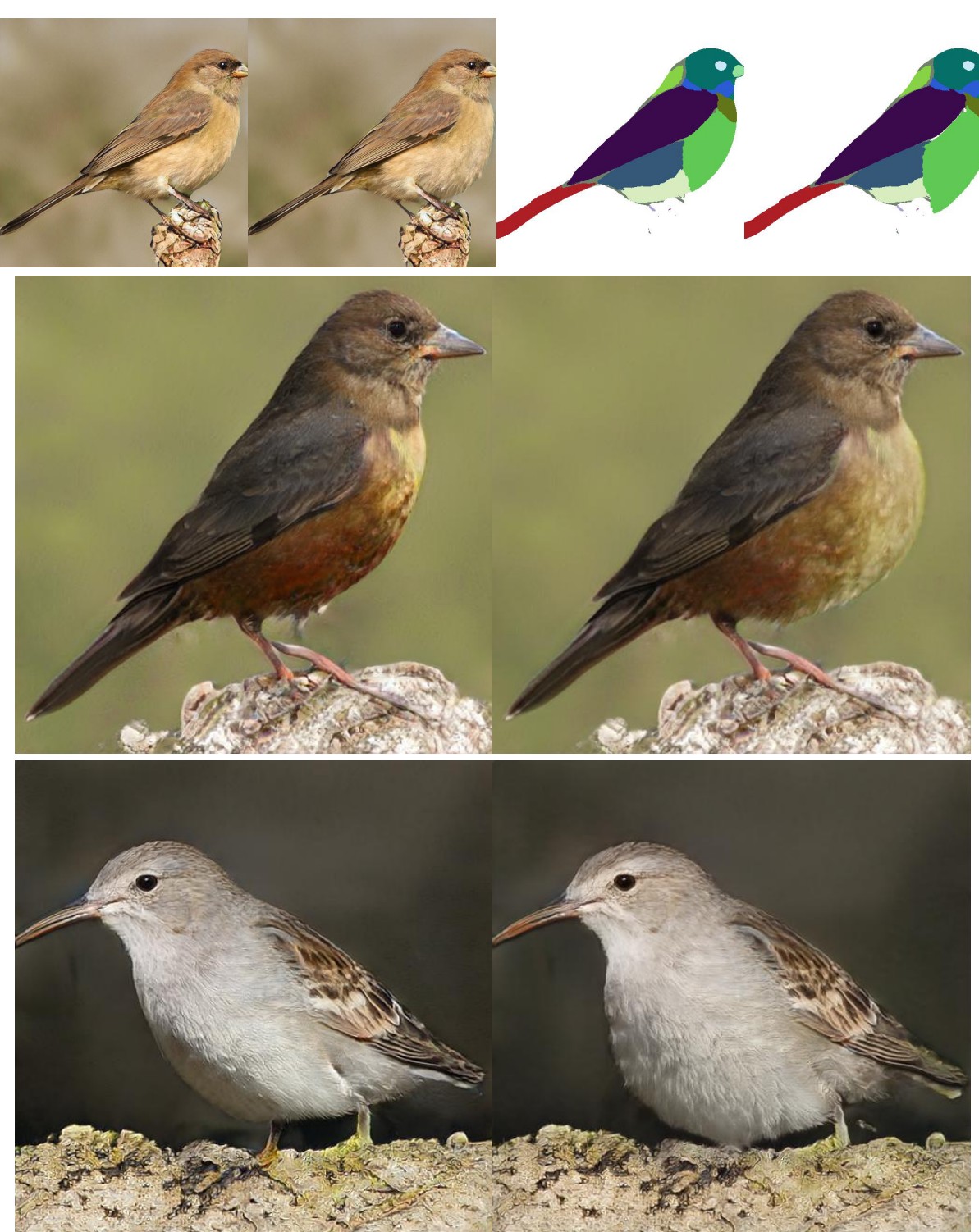

Figure 20: **Belly size editing.** *First row*: Image and mask pair to learn editing vector. Images are images before editing and after editing. Segmentation masks are before editing and target segmentation mask after manual modification. *Second and third rows*: Applying the learnt edit on new images.

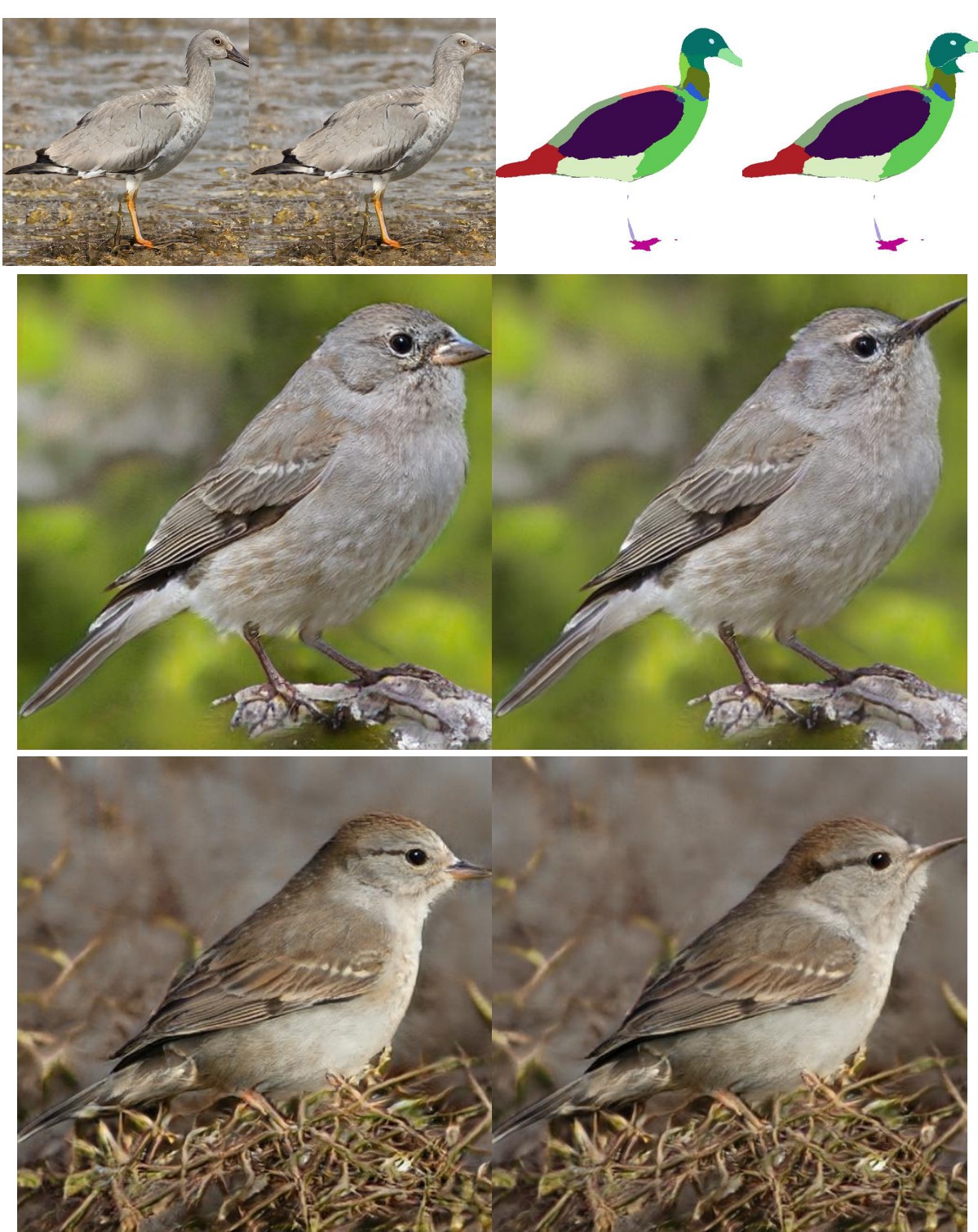

Figure 21: **Raising head editing.** *First row*: Image and mask pair to learn editing vector. Images are images before editing and after editing. Segmentation masks are before editing and target segmentation mask after manual modification. *Second and third rows*: Applying the learnt edit on new images.

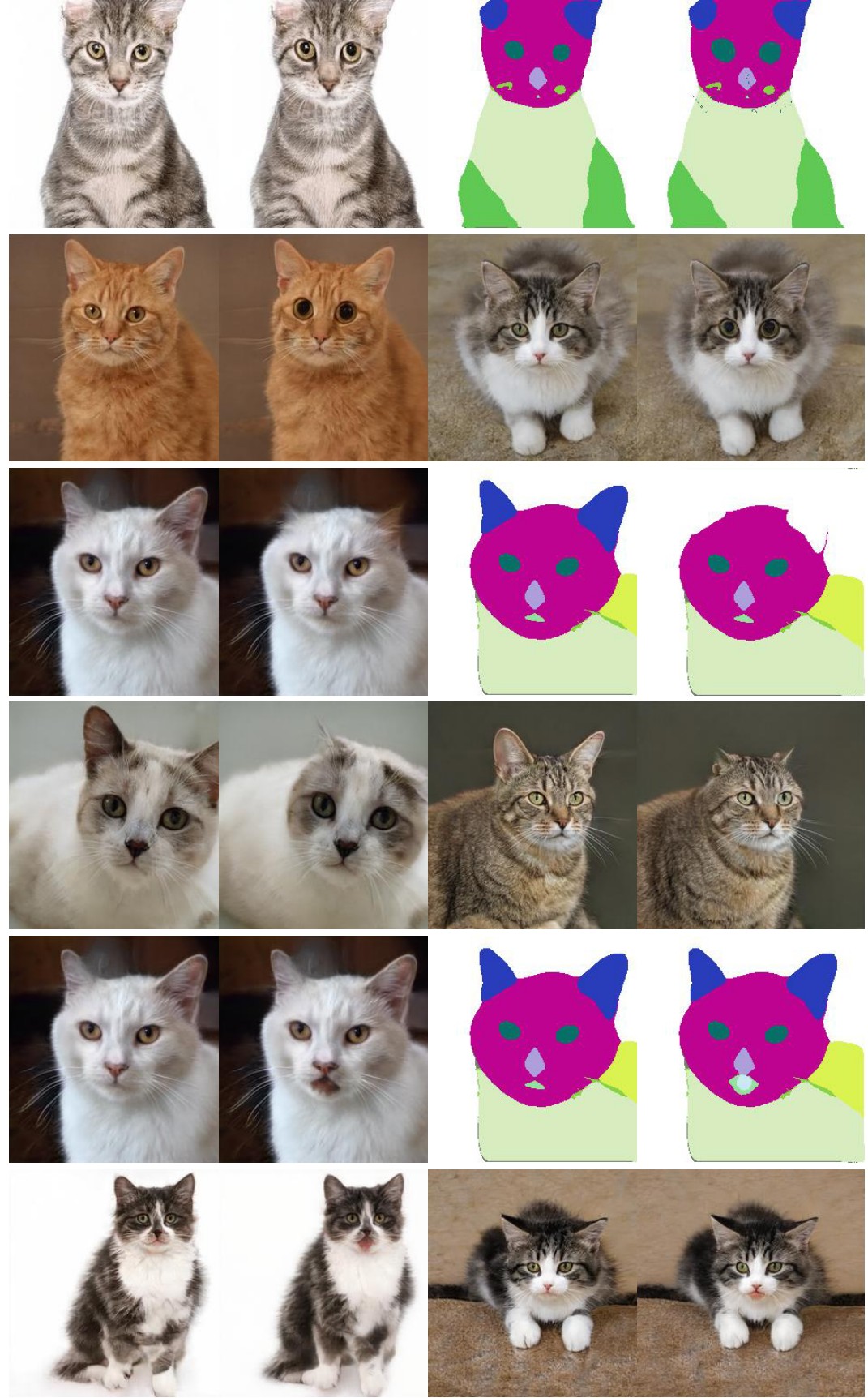

Figure 22: **Cat image editing.** *First and second rows*: **Eye size editing.** *Third and fourth rows*: **Ear size editing.** *Fifth and sixth rows*: **Open mouth editing.** *First, third, and fifth rows*: Image and mask pair to learn editing vector. Images are images before editing and after editing. Segmentation masks are before editing and target segmentation mask after manual modification. *Second, fourth, and sixth rows*: Applying the learnt edits on new images.