# OpenReview forum: "EditGAN: High-Precision Semantic Image Editing"
_NeurIPS.cc/2021/Conference — NeurIPS 2021 Poster_

### Official Review · Reviewer_7mk5 · 2021-06-30

**Rating:** 4
**Confidence:** 5

**Summary:**

This paper proposed a semantic image editing method that allows users to edit an input image by manually edit the corresponding parts of its predicted semantic segmentation map.

**Limitations And Societal Impact:**

Yes, the authors discussed both limitations and societal impact.
However, there is one missing limitation: the proposed method can only control the shape of the semantic editing regions but not their contents.

**Main Review:**

Overall, I think this paper is incremental and does not have enough contributions to be published at top venues:
1. This paper uses a combination of well-known methods: i) the joint modeling of images and their semantic segmentation [2][3] and ii) GAN inversion [4].
2. This is not the first paper that enables high-precision semantic image editing, e.g. SEAN [1]. In addition, methods like SEAN are more flexible as they enable users to control the styles in each semantic region while the proposed method cannot.
3. Compared to SEAN or similar methods, the major merit of the proposed method is that it uses less labeled data. However, this is a feature of previous studies [2][3].

So I think it is okay but not good enough.

[1] Peihao Zhu, Rameen Abdal, Yipeng Qin, and Peter Wonka. Sean: Image synthesis with semantic region adaptive normalization. In IEEE/CVF Conference on Computer Vision and Pattern Recognition (CVPR), June 2020.

[2] Yuxuan Zhang, Huan Ling, Jun Gao, Kangxue Yin, Jean-Francois Lafleche, Adela Barriuso, Antonio Torralba, and Sanja Fidler. Datasetgan: Efficient labeled data factory with minimal human effort. arXiv preprint arXiv:2104.06490, 2021.

[3] Daiqing Li, Junlin Yang, Karsten Kreis, Antonio Torralba, and Sanja Fidler. Semantic segmentation with generative models: Semi-supervised learning and strong out-of-domain generalization. arXiv preprint arXiv:2104.05833, 2021.

[4] Rameen Abdal, Yipeng Qin, and Peter Wonka. Image2stylegan: How to embed images into the stylegan latent space? In Proceedings of the IEEE International Conference on Computer Vision, pages 4432–4441, 2019.

**Originality**: This paper tackles an existing task using a new combination of existing methods (i.e. joint modeling of images and their semantic segmentation [2][3], GAN inversion [4]). As abovementioned, the contributions are incremental.

**Quality**: I think the methodology and claims of this paper are convincing as most of them are similar to those in [2][3] and [4]. The major problem is that it lacks new insights and is incremental.

**Clarity**: The submission is clearly written and easy to follow.

**Significance**: I think the results are okay but not good enough. As a combination of joint modeling of images and their semantic segmentation [2][3] and GAN inversion [4], the results are not surprising at all.


**Time Spent Reviewing:**

4

---

> ### Author Response · Authors · 2021-08-10
> **Authors' response**
>
> We would like to thank the reviewer for their constructive feedback.
>
> __Regarding novelty:__ We indeed build on top of DatasetGAN [1], SemanticGAN [2], and Image2stylegan [3]. However, this is merely the architecture that we use. Our editing protocol, our main contribution, is entirely novel. Importantly, the EditGAN procedure consists of multiple proposed steps:
> *(i)* Conditional optimization in latent space with a carefully chosen objective that optimizes for RGB pixel preservation outside the editing area of interest and for consistency with the edited segmentation inside the editing area (see Fig. 3). This approach translates the edit from segmentation to image space after modification of an image segmentation mask.
> *(ii)* Amortization of this optimization into learnt editing vectors, which translate to new images and can be applied in real time.
> *(iii)* Editing effect modulation at test time by scaling the editing vector with $s_\textrm{edit}$
> *(iv)* Self-supervised refinement to carefully control the edits when applying the editing vectors at test time to remove artifacts, if necessary. To the best of our knowledge such editing procedure has not been proposed before.
>
> Ultimately, our novel editing protocol leads to a framework with multiple advantages: (i) Possibility for real-time interactive editing; (ii) need for only very little annotated data and therefore broad applicability of the method in practice; (iii) unprecedented high-precision and detailed editing, while preserving full image quality (see, e.g., Fig. 7);(iv) possibility to easily combine multiple edits (see, e.g., Fig. 5).
> (v) working equally well on embedded real and GAN-generated images.
> To the best of our knowledge, no previous methods have all these advantages at once. We consider EditGAN as a simple and elegant editing method with excellent performance due to our carefully designed, novel editing protocol. We will try to better highlight these aspects in the final version of the paper.
>
> __Regarding Comparison to SEAN:__ Our method is significantly different from SEAN [4] in multiple important aspects.
> *(i)* SEAN directly conditions the generator on segmentation masks. Consequently, a very large amount of data with annotations is required for training. This severely limits the broad applicability of SEAN to images where such data is available. In their experiments, SEAN uses several thousands of segmentation annotations. This stands in stark contrast to EditGAN, where the segmentation mask is an additional output of the generator, similar to [1,2]. Most importantly, EditGAN requires only a handful of annotated data (16 in our case), which can be drawn relatively quickly and with the level of detail necessary for the desired edits.
> *(ii)* Related to the previous point, SEAN can not perform the kind of very detailed editing, which EditGAN can do. Consider, for example, the gaze change edits shown in Figure 4 or the pupil edits demonstrated in our Figure 7. SEAN would not be able to do such edits, as their employed segmentations (CelebAMask-HQ) do not have annotations of these details of the eyes. Obtaining such annotations in the scale necessary for training SEAN is expensive. This is precisely where our framework shines: since EditGAN requires only so few annotations we can quickly create annotations with an extreme level of detail for any dataset (see annotation schemas in Figure 6 of the main paper and Figure 1 of the Supplemental material).
> *(iii)* SEAN relies on reference style images. This can be a strength when such stylization is desired, but also problematic when no reference images are available. EditGAN does not rely on any reference images, which makes it generally broadly applicable, which was our goal when developing EditGAN. However, combining our EditGAN approach with style-based editing is an important and interesting future direction. Ultimately, our method primarily aims at very high-precision editing, where it outperforms all existing methods, and is complementary to methods that aim at stylization.
>
> Please let us know if you have further questions about EditGAN. We hope that we were able to eliminate your concerns about EditGAN and provide helpful answers. In that case, we would like to kindly ask you to consider adjusting the rating accordingly.
>
> [1] Yuxuan Zhang, Huan Ling, Jun Gao, Kangxue Yin, Jean-Francois Lafleche, Adela Barriuso, Antonio Torralba, and Sanja Fidler. Datasetgan: Efficient labeled data factory with minimal human effort. CVPR, 2021.
> [2] Daiqing Li, Junlin Yang, Karsten Kreis, Antonio Torralba, and Sanja Fidler. Semantic segmentation with generative models: Semi-supervised learning and strong out-of-domain generalization. CVPR, 2021.
> [3] Rameen Abdal, Yipeng Qin, and Peter Wonka. Image2stylegan: How to embed images into the stylegan latent space? ICCV, 2019.
> [4] Peihao Zhu, Rameen Abdal, Yipeng Qin, and Peter Wonka. Sean: Image synthesis with semantic region adaptive normalization. CVPR, 2020.

---

### Official Review · Reviewer_zqNn · 2021-07-13

**Rating:** 3
**Confidence:** 5

**Summary:**

The authors proposed a local image-editing algorithm based on pre-trained StyleGAN2. The motivation is simple. Editing is performed based on two branches: generation and parsing. Parsing relies on pre-defined segmentation masks. The incremental editing vector is learnt by constraints to edited images and associated parsing masks. To handle the disentanglement, additional optimization is needed to perform refinement.

**Limitations And Societal Impact:**

Yes

**Main Review:**

The paper is based on previous works: DatasetGAN and SemanticGAN. It can be viewed as a specific application of these two works on part editing.  But the improvement of the algorithm for image editing is nearly trivial. Besides, the similar idea is also applied in the following paper

Paint by word
https://arxiv.org/pdf/2103.10951.pdf

Replacing the NLP part with semantic masks in the above paper, the remaining core idea of solving editing vectors is nearly the same. Compared with Image2StyleGAN and Image2StyleGAN++, the improvement is the parsing branch, which is the idea of DatasetGAN and SemanticGAN. So, the novelty of the idea is too trivial compared with existing methods. I don't think the current state of the paper is suitable for NeurIPS publication.


%%%%%% comments after rebuttal %%%%%%

Thank you for your detailed feedback. I agree with the authors that they proposed a simple method for local image editing. But the limited novelty is obvious. I still think it is obviously below the bar of NeurIPS acceptance. We need to encourage innovation. You may revise your paper according to Reviewers' concerns, especially Reviewer M7u3 who gave constructive advice from several different aspects.

**Time Spent Reviewing:**

5 hours

---

> ### Author Response · Authors · 2021-08-10
> **Authors' response**
>
> We would like to thank the reviewer for their constructive feedback.
>
> __Regarding novelty:__ We indeed build on top of DatasetGAN and SemanticGAN. However, this is merely the architecture that we use. Our editing protocol, our main contribution, is entirely novel. Importantly, the EditGAN procedure consists of multiple proposed steps:
> (i) Conditional optimization in latent space with a carefully chosen objective that optimizes for RGB pixel preservation outside the editing area of interest and for consistency with the edited segmentation inside the editing area (see Fig. 3). This approach translates the edit from segmentation to image space after modification of an image segmentation mask.
> (ii) Amortization of this optimization into learnt editing vectors, which translate to new images and can be applied in real time.
> (iii) Editing effect modulation at test time by scaling the editing vector with $s_\textrm{edit}$
> (iv) Self-supervised refinement to carefully control the edits when applying the editing vectors at test time to remove artifacts. To the best of our knowledge such editing procedure has not been proposed before.
> Ultimately, our novel editing protocol leads to a framework with multiple advantages: (i) Possibility for real-time interactive editing; (ii) need for only very little annotated data and therefore broad applicability of the method in practice; (iii) unprecedented high-precision and detailed editing, while preserving full image quality (see, e.g., Fig. 7); (iv) possibility to easily combine multiple edits (see, e.g., Fig. 5). (v) working equally well on embedded real and GAN-generated images.
> To the best of our knowledge, no previous methods have all these advantages at once. We consider EditGAN as a simple and elegant editing method with excellent performance due to our carefully designed, novel editing protocol. We will try to better highlight these aspects in the final version of the paper.
>
> __Regarding “Paint by Word”:__ While “Paint by Word” shares some high-level similarities with EditGAN in the overall architecture, it differs in multiple important aspects:
> *(i)* “Paint by Word” relies on text-driven editing and leverages the CLIP model, which was trained using an extremely large database of image/caption pairs. Hence, “Paint by Word” implicitly relies on large-scale annotated data sets to realize the text-driven editing, contrary to EditGAN, which requires extremely little annotated data, segmentations in our case.
> *(ii)* “Paint by Word” seems to be purely optimization driven and does not allow for real-time editing, thereby severely limiting its usefulness in practice. In contrast, we propose to distill the optimization into editing vectors, which, as we demonstrate, directly translate to new images and allow us to perform complex and interactive real-time editing.
> *(iii)* “Paint by Word” has limited editing precision. All edits in their paper are relatively high-level scene modifications. It remains unclear whether very high-precision edits, like demonstrated by our framework, are possible in a purely text-driven manner.
> *(iv)* In summary, the two methods target different editing: While “Paint by Word” is all about text-driven editing, our goal is high-precision, real-time semantic editing. Consequently, we do not see “Paint by Word” as competitive, but as complementary to our EditGAN.
> Moreover, “Paint by Word” appeared on arXiv only recently, relatively shortly before the NeurIPS submission deadline, and can be considered concurrent with our method. All that being said, we thank the reviewer for pointing out this important work to us. We will include it into the related work discussion in the final version of our paper.
>
> Please let us know if you have further questions about EditGAN. We hope that we were able to eliminate your concerns about EditGAN and provide helpful answers. In that case, we would like to kindly ask you to consider adjusting the rating accordingly.

---

### Official Review · Reviewer_M7u3 · 2021-07-16

**Rating:** 5
**Confidence:** 5

**Summary:**

This work proposes EditGAN for semantic image editing with pre-trained GANs. It first introduces an additional branch beyond the original GAN generator to predict the semantic map corresponding to the image synthesis (this part is proposed by previous work). It then learns an encoder to project a target image to a latent code (this part is also proposed by previous work). At this end, given an image with the associated latent code and segmentation map, EditGAN enables semantic image editing by (1) modulating the segmentation map, (2) optimizing the latent code to fulfill the segmentation change, and (3) defining the semantic vector as the difference between the initial latent code and the optimized code. It turns out that such semantic vector is applicable to other images.


**Limitations And Societal Impact:**

The paper mentioned the broader impact in Sec.6.

**Main Review:**

Pros

- Good editing results, especially for the high-precision editing on details, such as spoke-rotation in Fig. 7.


Cons

- Limited novelty. The generator (StyleGAN2), segmentation branch (DatasetGAN), GAN inversion (pSp) are all proposed by previous work. Editing the segmentation map is able to edit the image correspondingly is not that surprising. From this perspective, the technical novelty is incremental.

- Lack of an important baseline. If I understand correctly, in the process of learning the edit vectors, users are required to modulate the segmentation map to make the modulation reasonable. For example, in Fig. 1, enlarging the wheels yet keeping the segmentation map reasonable is not that trivial. Then, a straightforward pipeline would be "what if directly changing the image, like using PhotoShop?". From my point of view, editing the image has the same difficulty as editing the segmentation map. In this way, we may not need the segmentation branch. More concretely, given an image A and an edited (with PhotoShop) image B, we can invert both of them to the latent space and get two latent codes, a and b. Then, (b-a) defines a semantic vector. According to my experience, such an operation is able to achieve wearing eyeglasses on face models. Without the comparison to such a baseline, I am not convinced that the proposed approach can give better results, but it requires much more effort than the aforementioned baseline.

- Limited editing flexibility.
(1) Is it possible to change the appearance of a particular segmentation part, e.g., changing the hair color?
(2) Is it possible to make the person one eye looking left and the other right?
(3) Enlarging ear and deleting ear in Fig. 4 do not make sense to me. Also, the ears are not fully deleted.
(4) When shrinking a certain part, how to fill the missing pixels seems to be a problem. Taking the first car on the 3rd row of Fig. 5 as an example, the missing parts are blurry after shrinking the wheels.


**Time Spent Reviewing:**

5

---

> ### Author Response · Authors · 2021-08-10
> **Authors' response**
>
> We would like to thank the reviewer for the constructive feedback.
>
> __Regarding novelty:__ We indeed build on top of StyleGAN2, DatasetGAN and the pSp encoder network. However, this is merely the architecture that we use. Our editing protocol, our main contribution, is entirely novel. Importantly, the EditGAN procedure consists of multiple proposed steps: (i) Conditional optimization in latent space with a carefully chosen objective that optimizes for RGB pixel preservation outside the editing area of interest and for consistency with the edited segmentation inside the editing area (see Fig. 3). This approach translates the edit from segmentation to image space after modification of an image segmentation mask. (ii) Amortization of this optimization into learnt editing vectors, which translate to new images and can be applied in real time. (iii) Editing effect modulation at test time by scaling the editing vector with $s_\textrm{edit}$ (iv) Self-supervised refinement to carefully control the edits when applying the editing vectors at test time to remove artifacts. To the best of our knowledge such editing procedure has not been proposed before. Ultimately, our novel editing protocol leads to a framework with multiple advantages: (i) Possibility for real-time interactive editing; (ii) need for only very little annotated data and therefore broad applicability of the method in practice; (iii) unprecedented high-precision and detailed editing, while preserving full image quality (see, e.g., Fig. 7); (iv) possibility to easily combine multiple edits (see, e.g., Fig. 5). (v) working equally well on embedded real and GAN-generated images.
>
> To the best of our knowledge, no previous methods have all these advantages at once. We consider EditGAN as a simple and elegant editing method with excellent performance due to our carefully designed, novel editing protocol. We will try to better highlight these aspects in the final version of the paper.
>
> __Regarding Photoshop baseline:__ We respectfully disagree about this point. Manually editing segmentation masks is usually very easy and boils down to painting and scaling objects. A successful and well-known segmentation editing tool is GauGAN as presented in [1] and we envision a similar interface for EditGAN. The reviewer pointed out the wheel size edit specifically: Fig. 16 in the supplemental material shows the segmentation modification that was used to learn this edit. It took us only a few seconds to quickly select and scale the wheel segmentation elements. On the other hand, directly editing images on the RGB pixel level is significantly harder: We would have to preserve photorealism, plausible lighting conditions and shadows, and we may have to insert new details. Examples where this becomes particularly clear are the smile edit (we would have to manually draw additional teeth that become visible, which is difficult) or the large-scale edit where we convert the car type (Fig. 8 in main paper; here, we would have to draw the whole back of the car from scratch in a photorealistic fashion). The segmentation mask modifications for these edits only take a few seconds.
> We could also use copy-pasting and distortion methods in tools such as Photoshop to realize various image modifications. However, this usually comes with strong non-photorealistic artifacts, which would then be encoded in the learnt editing vectors. This would be very problematic and strongly degrade editing performance. A skilled digital artist may be able to create plausible photorealistic edits directly in Photoshop, given enough time. However, the promise of tools like EditGAN---and many other methods developed in the AI-driven image editing literature---is to democratize high quality image editing and make it easy and efficient for the average, non-professional user. With EditGAN, the user only needs to coarsely modify segmentation masks and/or apply fixed editing vectors.
> Note that we are currently in the process of developing an interactive tool which we are planning to release with the paper. Users can then directly try out EditGAN on their images and play with different editing vectors and learn their own editing vectors.
>
> __Regarding Editing Flexibility:__
> *Reply to (1):* Changing the appearance, such as color or texture, of individual segmentation parts is currently not possible and out of the scope of our method, which focuses on highly detailed as well as large-scale semantic and shape modifications. However, we could potentially include such information in the annotation scheme and extend EditGAN accordingly or combine EditGAN with other editing methods that allow for such edits. This is a highly interesting direction for future research. As another potential direction towards that, it would be interesting to leverage high quality synthetic 3D models, which can be rendered with different colors, textures or under different lighting conditions to find corresponding editing vectors in latent space using our framework for finding these vectors.
> *Reply to (2):* Yes, we can also change the gaze such that one eye points to the right and the other to the left. We were hesitant to include extreme or unrealistic face edits, since faces are sensitive objects, but we can potentially include the suggested edit in the final version of the paper.
> *Reply to (3) and (4):* While we do not see any problems with the ear size modification edit, it needs to be pointed out that edits can introduce artifacts when we push them to the extreme. We need to keep in mind that the GAN has never seen images of cats without or overly large ears. Similarly, the GAN has never seen cars without wheels and hence won’t know how to introduce meaningful details when the edit is applied with such a strong magnitude that the wheels almost disappear entirely. The visible artifacts in that case are expected behavior and simply indicate that the edit has been applied with a magnitude that is likely too strong. We would like to thank the reviewer for bringing our attention to this and we will discuss this aspect in more detail in the final version of the paper.
>
> We hope that we were able to clarify some aspects of EditGAN, better highlight our contributions, and answer the questions. Please let us know if you have further questions or concerns. We would like to kindly ask you to consider adjusting the rating accordingly, if our replies were satisfactory.
>
> [1] http://nvidia-research-mingyuliu.com/gaugan/

---

### Official Review · Reviewer_9mLQ · 2021-07-16

**Rating:** 5
**Confidence:** 4

**Summary:**

The paper proposes a new method for semantic image editing, which allows users to edit images by changing their segmentation masks. More specifically, the method embeds an image required to be modified into the GAN's latent space and then perform optimisation according to the segmentation edit.

**Limitations And Societal Impact:**

Limitations:
1. The proposed editing method relies heavily on the shared latent code between images and their segmentation masks. However, image and segmentation masks are different modal information, so how do authors ensure these two features are exactly match in latent space?

2. The method depends on the quality of segmentation masks, so if the generated masks fail to separate each part of an object, can proposed method still modify desired regions? e.g., in cat and bird examples, the proposed method seems to have limited modification choices.

3. Can proposed method control the scale of modified effect?

4. The proposed method builds on top of DatasetGAN, relying on modifying generated segmentation masks. Although authors show some differences in line 102, it might be better to discuss more about their differences or highlight their contribution.




**Main Review:**

Originality: The proposed method builds on DatasetGAN and SemanticGAN, but allows users to achieve image editing. The difference between proposed method and previous works have been shown in related work.

Quality: The submission is technically sound, claims are well supported, and authors have discussed strengths and weaknesses of the proposed method.

Clarity: The paper is well organised and easily followed.

Significance: The results look good, which achieve high-quality image editing.



**Time Spent Reviewing:**

8

---

> ### Author Response · Authors · 2021-08-10
> **Authors' response**
>
> We would like to thank the reviewer for the constructive feedback.
>
> __Regarding matching images and semantic segmentation masks in latent space:__ To match the segmentation masks and images in latent space, we build on DatasetGAN [1]. In short, given a trained StyleGAN, we connect a separate segmentation branch to the feature maps of the StyleGAN generator. To train this segmentation branch, we take labeled images and embed them into the GAN’s latent space. After embedding, we have latent codes that generate the labeled images (and corresponding feature maps along the way). Now, we can train the segmentation branch to also generate the corresponding segmentation annotation from the same latent code. That way, we learn a generator that jointly generates images and their segmentation masks from the *same* underlying latent code.
> In the interest of space, this was not discussed in detail in the main paper, where we focus on our novel editing method. The training of the StyleGAN generator, the encoder, and the segmentation branch are explained in detail in the supplementary material.
> Importantly, learning this segmentation branch is done in a separate step after learning the main StyleGAN for image generation and requires only a tiny amount of annotated training images (16 annotated images are sufficient). This makes our method easily applicable on diverse data sets and much more broadly applicable than other editing methods that often require large annotated data sets. 16 annotations can always be created very quickly and the choice of the segmentation schema can even be tailored to the edits the user envisions. This is why we were able to demonstrate editing on multiple different kinds of images (faces, cars, birds, cats) and with very high precision, while the majority of the editing literature often only demonstrates editing of face images, where richly annotated data sets exist.
>
> __Regarding EditGAN’s reliance on the quality of the segmentation masks:__ This is closely related to the previous point and there are two aspects.
> *(i):* Our editing capabilities are determined and limited by the level of detail of the segmentation annotations. As discussed, since we only require a handful of annotations, we can opt for creating highly detailed segmentations. The segmentation schema for faces is shown in Figure 6 of the main paper, while the segmentation schemas for cat, bird and car are shown in Figure 1 of the supplementary material. In the paper, we demonstrated only a few possible editing possibilities, but we can easily create many more different edits, leveraging the detailed segmentation schemes. See, for example, Figure 7 for highly detailed editing. Many more edits are shown in the supplementary material; see Figures 18-21 for editing examples of bird and cat images.
> *(ii):* When inferring a segmentation mask with the segmentation branch, we rely on this branch’s performance. In general, we find the segmentation branch’s accuracy to be very reliable. More importantly, when learning an editing vector---which needs to be done only once---we can pick an image for which the predicted segmentation is highly accurate. Afterwards, the editing vector can be applied on new images without explicitly relying on the segmentation branch again and generate reliable edits independently of the prediction of the segmentation branch.
>
> __Regarding scaling of the editing effect:__ Yes, when applying editing vectors in latent space we can scale the vector and thereby modulate the effect. This is discussed in section in 3.4 of the manuscript, where the scaling coefficient is defined as $s_\textrm{edit}$. It is experimentally demonstrated for example in Figure 9 in the main paper and Figure 3 in supplementary material..
>
> __Regarding connections to DatasetGAN:__ Our method indeed builds on top of the DatasetGAN model. In DatasetGAN [1], the generator that jointly generates images and their segmentations is used to synthesize a large simulated data set, which is used to train downstream segmentation models. We merely leverage DatasetGAN as the backbone for our novel EditGAN technique. We demonstrate that a GAN that *jointly* models images and their segmentations can be leveraged for high-precision semantic image editing. To this end, we propose a novel editing framework that consists of multiple aspects: (i) Conditional optimization in latent space after modification of an image segmentation mask. (ii) Amortization of this optimization into learnt editing vectors, which translate to new images and can be applied in real time. (iii) Editing effect scaling and self-supervised refinement techniques to carefully control the edits when applying the editing vectors at test time. In summary, our editing framework, *EditGAN*, allows very high-precision semantic image editing, in real-time, and on diverse images, both real and GAN-generated, while relying on very few annotations. We will try to better highlight these technical contributions and advantages of our framework in the final paper version.
>
> Please let us know if we were able to answer the questions and eliminate concerns about EditGAN. In case our reply is satisfactory, we would like to kindly ask the reviewer to consider adapting the paper rating accordingly. Thank you very much!
>
> [1] Zhang et al. DatasetGAN: Efficient Labeled Data Factory with Minimal Human Effort. CVPR, 2021.

---

### Author Response · Authors · 2021-08-10
**Additional general reply to all reviewers**

We would like to thank all reviewers for their thoughtful feedback on EditGAN.
One common point of concern seems to be around our technical novelty and our advantages over existing editing methods. As we evidently did not communicate this sufficiently well in the manuscript, we would like to briefly address this here to clarify. We will improve the presentation in the final version of the paper.

Regarding our backbone model, EditGAN builds on top of DatasetGAN [1], SemanticGAN [2], and Image2stylegan [3]. However, this is merely the architecture that we use. Our editing protocol, our main contribution, is entirely novel. Importantly, the EditGAN procedure consists of multiple proposed steps:

1. Conditional optimization in latent space with a carefully chosen objective that optimizes for RGB pixel preservation outside the editing area of interest and for consistency with the edited segmentation inside the editing area (see Fig. 3). This approach translates the edit from segmentation to image space after modification of an image segmentation mask.
2. Amortization of this optimization into learnt editing vectors, which directly translate to new images and can be applied in real time.
3. Editing effect modulation at test time by scaling the editing vector with $s_\textrm{edit}$
4. Optional self-supervised refinement to carefully control the edits when applying the editing vectors at test time to remove artifacts, if necessary.

To the best of our knowledge such an editing protocol has not been proposed before. However, it is crucial for EditGAN’s strong performance. Ultimately, our novel editing procedure leads to a framework with multiple advantages:

1. Possibility for real-time interactive editing with optional refinement.
2. Need for only very little annotated data and therefore broad applicability to various image types, for which no large annotated data sets exist.
3. State-of-the-art high-precision and detailed editing, while preserving full image quality.
4. Possibility to easily combine multiple edits.
5. EditGAN works equally well on embedded real and GAN-generated images.

To the best of our knowledge, no previous methods have all these advantages at once. We consider EditGAN as a simple and elegant editing method with excellent performance due to our carefully designed, novel editing protocol. We will try to better highlight these aspects in the final version of the paper.

*[1] Yuxuan Zhang, Huan Ling, Jun Gao, Kangxue Yin, Jean-Francois Lafleche, Adela Barriuso, Antonio Torralba, and Sanja Fidler.* Datasetgan: Efficient labeled data factory with minimal human effort. *CVPR, 2021.*

*[2] Daiqing Li, Junlin Yang, Karsten Kreis, Antonio Torralba, and Sanja Fidler.* Semantic segmentation with generative models: Semi-supervised learning and strong out-of-domain generalization. *CVPR, 2021.*

*[3] Rameen Abdal, Yipeng Qin, and Peter Wonka.* Image2stylegan: How to embed images into the stylegan latent space? *ICCV, 2019.*

---

### Decision · Program_Chairs · 2021-09-28

**Decision:**

Accept (Poster)

**Comment:**

The paper proposes a new method for GAN-based image editing with the pipeline where a user modifies a segmentation mask of an image. While the paper is well-written and the editing results are impressive, four reviewers unanimously consider the technical novelty of the paper as limited and the paper being incremental. Since none of the reviewers advocated for this work, my recommendation is "Reject".

**Consistency Experiment:**

NeurIPS has a long history of experimentation. In 2014, NeurIPS ran an experiment in which 10% of submissions were reviewed by two independent committees to quantify the randomness in the review process. This year, we repeated a variant of this experiment to see how the quality of the review process has changed over time.  This paper was part of the experiment and was therefore assigned to two committees (consisting of reviewers, an Area Chair, and a Senior Area Chair) that reached independent decisions.  If both committees made the same recommendation, this recommendation was followed. If a single committee recommended acceptance, the paper was accepted (with the exception of a few cases in which the other committee identified what we considered a fatal flaw, e.g., an error in a key result).

This copy’s committee reached the following decision: **Reject**

The other committee assigned to the paper recommended **Accept (Poster)**.  You can find the other set of reviews, along with any follow up discussion with the authors here:
https://openreview.net/forum?id=ppv5yqhpNyE